



# Grain growth of natural and synthetic ice at 0ºC

Sheng Fan[1,2], David J. Prior[2], Brent Pooley[2], Hamish Bowman[2], Lucy Davidson[2], Sandra Piazolo[3], Chao Qi[4,5], David L. Goldsby[6], Travis F. Hager[6]

[1] Department of Earth Sciences, University of Cambridge, Cambridge, UK
[2] Department of Geology, University of Otago, Dunedin, New Zealand
[3] School of Earth and Environment, University of Leeds, Leeds, UK
[4] Key Laboratory of Earth and Planetary Physics, Institute of Geology and Geophysics, Chinese Academy of Sciences, Beijing, China
[5] College of Earth and Planetary Sciences, University of Chinese Academy of Sciences, Beijing, China
[6] Department of Earth and Environmental Science, University of Pennsylvania, Philadelphia, PA, USA

*Correspondence to:* Sheng Fan (sf726@cam.ac.uk).

## Abstract

Grain growth can modify the microstructure of natural ice, including the grain size and crystallographic preferred orientation (CPO). To understand better grain-growth processes and kinetics, we compared microstructural data from synthetic and natural
ice samples that were annealed at ice-solidus temperature (0ºC) to successfully long durations. The synthetic ice has a homogeneous initial microstructure, which is characterised by polygonal grains, little intragranular distortion and bubble content, and a near-random CPO. The natural ice samples were sub-sampled from ice cores acquired from the Priestley Glacier, Antarctica; they have a heterogeneous microstructure, which is characterised by a considerable number of air bubbles, widespread intragranular distortion, and a preferred crystallographic alignment. During annealing, the average grain size of
natural ice barely changes, whilst the average grain size of synthetic ice gradually increases. This observation suggests grain growth in natural ice can be much slower than synthetic ice; the grain-growth law derived from synthetic ice data cannot be directly applied to estimate the grain-size evolution in natural ice. The microstructure of natural ice characterised by many bubbles pinning at grain boundaries. Previous studies suggest bubble pinning reduces the driving force of grain boundary migration, and it should be directly linked to an inhibition of grain growth observed in natural ice. As annealing progresses,
the number density (number per unit area) of bubbles on natural-ice grain boundaries decreases, whilst the number density of bubbles in grain interior increases. This observation indicates that some ice grain boundaries sweep through bubbles, which should weaken the bubble-pinning effect and thus enhance the driving force for grain boundary migration. Consequently, the grain growth in natural ice might comprise more than one stage and it should correspond to more than one set of grain-growth parameters. Some of the Priestley ice grains become abnormally large during annealing. We suggest the bubble-pinning, which
inhibits the grain growth of ice matrix, and the contrast of dislocation-density amongst neighbouring grains, which favours the selected growth of individual grains with low dislocation densities, are tightly correlated with the abnormal grain growth.



## 1 Introduction

The mechanical behaviour of polycrystalline materials, such as metals, rock-forming minerals, and ice, is largely dependent
on the microstructure, e.g., grain size and crystallographic preferred orientation (CPO), of crystalline grains (Langdon, 1970;
Warren and Hirth, 2006; Goldsby and Kohlstedt, 1997, 2001; Dillamore et al., 1979; Azuma, 1995). Previous ice deformation
experiments show: (1) Ice rheological behaviour is grain-size sensitive. Samples with smaller grain sizes are mechanically
weaker, i.e., exhibiting lower stresses under constant displacement rates or higher strain rates under constant loads, compared
with samples with larger grain sizes (Goldsby and Kohlstedt, 1997; Qi and Goldsby, 2021). (2) Ice crystals exhibit a strong
viscoplastic anisotropy. Ice samples with the majority of grains orienting for easy slip, e.g., with basal planes at 45° to the
maximum principal stress, are mechanically much weaker (~60 times weaker for a single crystal) compared to samples with
the majority of grains orienting for hard slip, e.g., with basal planes at 0° or 90° to the maximum principal stress (Azuma, 1995;
Duval et al., 1983; Lile, 1978). Ice deformation flow laws (e.g., Glen, 1955) quantify the relationship between stress and strain
rate; some flow laws include the effects of grain size (Goldsby and Kohlstedt, 1997, 2001) and CPO (Azuma, 1994, 1995).
Many ice-core studies use ice grain size and/or CPO statistics to estimate the deformation history of ice flows (Montagnat et
al., 2014; Kuiper et al., 2020a, b; Weikusat et al., 2017b; Thomas et al., 2021; Gerbi et al., 2021). Recent experimental and
modelling work further emphasized the importance of incorporating grain size and CPO in the modelling of ice-sheet dynamics
(Fan et al., 2021a; Behn et al., 2021; Llorens et al., 2022; Rathmann and Lilien, 2021; Kuiper et al., 2020a).

Grain growth inferred from an increasing grain size with the elapsed-time/depth of burial in the shallow parts of polar ice
cores, is an important process in natural ice (Duval, 1985; Alley et al., 1986; Thorsteinsson et al., 1997; Gow and Williamson,
1976; Duval and Lorius, 1980). Grain growth within a single-phase system is considered a result of normal grain growth,
where boundary energy drives the movement (migration) of grain boundaries to reduce the total boundary surface area
(Atkinson, 1988). Laboratory experiments show, for synthetic starting materials, normal grain growth will "continuously"
increase the average grain size (Karato, 1989; Evans et al., 2001; Faul and Scott, 2006; Azuma et al., 2012).

The microstructure of natural ice is largely different from synthetic ice.

(1) Impurities. Synthetic ice, which is usually made from ultra-pure water, contains few impurities (Cole, 1979). On
the contrary, natural ice is impurity-rich; usually containing insoluble (e.g., dust particles) and soluble (e.g., dissolved salts)
impurities, as well as air bubbles (Gow, 1968; Gow and Williamson, 1971; Svensson et al., 2005; Faria et al., 2010; Thomas
et al., 2021; Baker et al., 2003; Weikusat et al., 2017b).

   (2) Grain structure and CPO. Synthetic ice usually has a uniform microstructure characterised by straight or slightly
curved grain boundaries, polygonal grain shape, minimised intragranular distortions, and a random CPO (Cole, 1979; Fan et
al., 2020). On the contrary, natural ice contains a more complex microstructure that can include irregular grain boundaries,
shaped grains, considerable intragranular distortions, and a preferred alignment of ice crystalline axes (Weikusat et al., 2017a;
Thomas et al., 2021).



Microstructural features have a strong impact on grain growth. For example, bubbles and particles will introduce a dragging

force that can pin grain boundaries, reducing the driving force for boundary movement and the rate of grain growth (Azuma et al., 2012; Roessiger et al., 2014; Herwegh et al., 2011). Strain-energy stored within crystal lattice will promote nucleation, which produces small grains at the cost of original grains and thus slows down grain growth (Wilson, 1982; Piazolo et al., 2006). Consequently, the grain growth rate in natural ice, where secondary phases and anisotropic microstructures are a norm, should be different from the growth rate predicted from idealised synthetic ice. However, our understanding of grain growth

in natural ice is limited, because nearly all the experiment or modelling based grain-growth data are from synthetic ice (e.g., Azuma et al., 2012; Roessiger et al., 2014).

During sampling, transportation, and storage of natural ice cores, temperature changes might introduce a modification of ice microstructure. For example, during hot-water drilling, the circulation of pressurized hot water, which is used to produce and maintain the opening of boreholes, can increase the temperature of the ice core up to 0ºC (Humphrey and Echelmeyer, 1990;

Kamb, 2001; Jackson, 1999; Jackson and Kamb, 1997; Taylor, 1984). Moreover, the back-flowing of subglacial water into the borehole, which usually happens during the drilling of deep ice cores (Oerter et al., 2009), might also drastically increase the temperature of ice cores. Further, it may take days to acquire ice cores from the water-filled borehole (at ~0ºC), if the drilling meets technical difficulties such as borehole refreezing (Humphrey and Echelmeyer, 1990). Consequently, it is important to assess the role of abrupt temperature changes on the modification of natural-ice microstructure so that we can evaluate the

microstructural changes that might be introduced during the collection of natural ice samples.

In this contribution, we present detailed microscopic analyses on synthetic pure-water ice and natural ice samples that were annealed at ice-solidus temperature (0ºC). The goal of this study is to assess the effects of initial ice microstructures on grain-growth processes and kinetics. Particularly, we aim at better understanding the grain growth in natural ice.

## 2. Methods

### 2.1 Sample fabrication

We used two types of samples with a similar starting grain size of ~900 µm for annealing experiments: (1) medium-grained synthetic ice, and (2) natural ice collected from a fast-shearing margin of the Priestley Glacier, Antarctica.

Polycrystalline synthetic ice was fabricated using a "flood-freeze" method (Cole, 1979; Stern et al., 1997). Firstly, ice seeds made from frozen ultra-pure, deionized water were sieved between mesh sizes of 1 and 2 mm. A "wet sieve" method (Fan et

al., 2021b), i.e., pouring liquid nitrogen over ice seeds while sieving, was applied to reduce the number of finer grains that electrostatically clump on the surface of coarser grains. After that, the sieved ice seeds were packed into greased cylindrical moulds with inner diameters of 25.4 or 40 mm. The packed moulds were evacuated to remove air from void space and thermally equilibrated within a water-ice bath at 0°C for 30 minutes before they were flooded with 0°C ultra-pure, deionized water. Soon after, the flooded moulds were placed vertically within a chest freezer at ~-30°C with the bases of moulds directly contacting

a metal plate and with the walls of moulds insulated by a polystyrene block. This process ensures that the frozen front migrates



from the bottom of the moulds and slowly upwards so that the gas bubbles can be eliminated. After ~24 hours, ice cores were gently pushed out from the moulds using an arbor press. Each synthetic ice core was cut normal to the cylindrical axis into several cylindrical ice slabs using a bandsaw (Fig. 1(a)). These ice slabs will be used for annealing experiments. The thickness for ice slabs with a diameter of 25.4 mm is 15 mm; the thickness for ice labs with a diameter of 40 mm is 20 mm (Fig. 1(a)).

Natural ice samples were subsampled with a bandsaw from an Antarctic ice core (core no. 30) collected at a depth of ~26 m from a fast-shearing margin of the Priestley Glacier (Thomas et al., 2021). We first cut the ice core (diameter of ~105 mm) in half along its cylindrical axis. Ice disks, with a thickness of ~20 mm, were produced by sectioning perpendicular to the long axis of one of the half-cylindrical ice cores (Fig. 1(a)). From each ice disk, we produced two cuboid ice slabs with a dimension of ~35 (length) × 30 (width) × 20 (height) mm (Fig. (a)). The top and bottom surfaces of the cuboid ice slabs correspond to

the macroscopic profile plane (i.e., the plane normal to the shear plane) of the ice flow (Fig. 1(a)).





**Figure 1. (a) Schematic drawing for subsampling ice slabs from synthetic and natural ice cores. (b) Schematic drawing for surfaces selected and prepared from ice slabs with different dimensions for EBSD data collection.**



## 2.2 Annealing experiments

Annealing experiments at ~0°C were carried out at the Ice Lab, University of Otago. Two types of rigs (Rig01, Rig02; Figs 2(a–b)) were used for the experiments with minimized temperature fluctuation.

Figure 2(a) illustrates the design of Rig01, which was used for long-term annealing experiments (up to a month). Rig01 has two aluminium sample bins submerged in water-ice mixture in an insulated box (chilly bin). The chilly bin sits in a fridge maintained at -2 to 4°C. The sample bins are filled with silicon oil and fixed beneath two PVC bars using cable ties. The PVC

bars are tightly mounted on the walls of the chilly bin by friction to overcome the water buoyancy that pushes bins upwards. Before experiments, ice slabs were vacuum-sealed in plastic bags using a food vacuum sealer at ~-30°C in a chest freezer. This process was conducted to physically isolate ice samples from silicon oil as the contamination of silicon oil on sample surface will bring problems to electron backscatter diffraction (EBSD) data collection. When starting the experiments, bags of ice slabs were quickly transferred to sample bins and submerged within silicon oil at 0°C, within 30 seconds. For each sample bin,

a bag of gravel (weights of ~300 g; kept at ~-30°C) was then gently placed on the top of ice slabs to prevent them from floating. After that, the openings of sample bins and the PVC chilly bin were insulated with layers of wool pads sealed in plastic bags. Temperatures of the silicon oil bath within each of the sample bins and the temperature of the water-ice bath were recorded once every two seconds throughout the experiments (Fig. 2(c)). During experiments, we maintained the temperature of silicon oil at ~0°C by recharging the water-ice bath once every seven days. We did this by simultaneously removing extra water via a

drain valve at the base of the chilly bin and refilling ice cubes through the opening of the chilly bin; this process took 5 minutes. The upper level of the water-ice bath (within the chilly bin) was kept similar to or higher than the upper level of silicon oil (within the sample bins) throughout experiments. These processes ensure an insignificant temperature change within the silicon oil even during a maximum experimental time of one month (Fig. 2(c)).

Figure 2(b) illustrates the design of Rig02, which was used for short-term annealing experiments (up to 5 days). Rig02 is

composed of a thick-walled polystyrene box insulated by layers of sealed wool pads. The box sits in a room at ~20°C. Before each experiment, the polystyrene box was filled with a water-ice mixture. Ice slabs were sealed within cylindrical aluminium vessels with an inner diameter of ~26 mm, and the assemblies were kept within a freezer at ~-30°C. One of the ice-vessel assemblies had a thermometer directly frozen into the ice slab and it was used for measuring reference ice temperature during experiments. When starting the experiments, ice-vessel assemblies, including the reference sample, were transferred into the

polystyrene box and submerged in the water-ice bath within 30 seconds. Aluminium vessels were in direct contact with a metal plate placed at the bottom of the polystyrene box. After that, we insulated the opening of the polystyrene box with a polystyrene cap and wool pads. Temperatures of the reference ice sample and water-ice bath were recorded once every two seconds throughout the experiments (Fig. 2(d)). During experiments, we stabilized the temperature of the ice sample at ~0°C by recharging the water-ice bath once every 8 hours.



After annealing, bags or vessels containing ice slabs were removed from rigs (marked with arrows in Figs. 2(c–d); details in Table 1). Ice slabs were immediately removed from bags or vessels within a chest freezer at ~-30℃. These samples were then progressively cooled to ~-30, -100, and -196℃ within 15 minutes and thereafter stored in a liquid nitrogen dewar.

**Table 1 Details of ice annealing experiments**

| Annealing time (hours) | Sample type | Sample number | Initial median ice grain size (μm) | Measured ice grain size after annealing (μm) (lower quartile/medium/higher quartile) | | Number of ice grains measured after annealing | |
|---|---|---|---|---|---|---|---|
| | | | | Individual section | Combined sections | Individual section | Combined sections |
| 24.75 | Synthetic | 1_S_M_A | 907 | 853/**1284**/1776 | 820/**1228**/1652 | 514 | 866 |
| | | 1_S_M_B | | 766/**1187**/1500 | | 352 | |
| 49.65 | Synthetic | 2_S_M_A | 907 | 884/**1303**/1750 | 941/**1386**/1893 | 446 | 821 |
| | | 2_S_M_B | | 1017/**1512**/2088 | | 375 | |
| 96.00 | Synthetic | 3_S_M_A | 907 | 1472/**2351**/3225 | 1010/**1675**/2438 | 56 | 264 |
| | | 3_S_M_B | | 929/**1517**/2298 | | 208 | |
| 194.25 | Synthetic | 4_S_M_A | 907 | 1426/**2120**/2968 | 1334/**2138**/3065 | 120 | 192 |
| | | 4_S_M_B | | 1185/**2171**/3150 | | 72 | |
| 433.93 | Synthetic | 5_S_M_A | 907 | 1338/**2567**/4306 | 1338/**2567**/4306 | 89 | 89 |
| | | 5_S_M_B* | | N/A | | N/A | |
| 648.03 | Synthetic | 6_S_M_A | 907 | 1571/**3275**/4853 | 1571/**3275**/4853 | 48 | 48 |
| | | 6_S_M_B* | | N/A | | N/A | |
| 1.98 | Natural (Priestley glacier) | 7_P_A | 917 | 546/**1008**/1795 | 574/**1009**/1649 | 460 | 1110 |
| | | 7_P_B | | 595/**1009**/1565 | | 650 | |
| 3.98 | Natural (Priestley glacier) | 8_P_A | 917 | 556/**1029**/1566 | 559/**1022**/1650 | 707 | 1345 |
| | | 8_P_B | | 562/**1020**/1742 | | 638 | |
| 8.27 | Natural (Priestley glacier) | 9_P_A | 917 | 603/**1039**/1681 | 600/**1070**/1731 | 674 | 1230 |
| | | 9_P_B | | 560/**1099**/1810 | | 556 | |
| 24.05 | Natural (Priestley glacier) | 10_P_A | 917 | 603/**1061**/1822 | 569/**1063**/1788 | 663 | 1127 |
| | | 10_P_B | | 529/**1070**/1727 | | 464 | |
| 72.10 | Natural (Priestley glacier) | 11_P_A | 917 | 503/**999**/1764 | 555/**1034**/1708 | 432 | 896 |
| | | 11_P_B | | 600/**1049**/1682 | | 464 | |
| 174.37 | Natural (Priestley glacier) | 12_P_A | 917 | 513/**996**/1588 | 479/**956**/1571 | N/A | 1539 |
| | | 12_P_B | | 452/**932**/1562 | | 171 | |
| 360.20 | | 13_P_A | 917 | 430/**928**/1524 | 479/**972**/1577 | 682 | 1309 |





| | Natural (Priestley glacier) | **13_P_B** | | 528/**997**/1579 | | 857 | |
|---|---|---|---|---|---|---|---|
| 304.43 | Synthetic | **14_S_M_A** | 907 | 1425/**2521**/3896 | 1421/**2617**/3860 | 54 | 116 |
| | | **14_S_M_B** | | 1453/**2617**/3818 | | 62 | |
| 797.67 | Natural (Priestley glacier) | **15_P_A*** | 917 | N/A | 393/**1100**/2234 | N/A | 171 |
| | | **15_P_B** | | 393/**1100**/2234 | | 171 | |
| 1.00 | Synthetic | **20_S_M** | 907 | 553/**1008**/1412 | N/A | 367 | N/A |
| 2.00 | Synthetic | **21_S_M** | 907 | 659/**1072**/1449 | N/A | 337 | N/A |
| 4.00 | Synthetic | **22_S_M** | 907 | 524/**1032**/1580 | N/A | 330 | N/A |
| 8.00 | Synthetic | **23_S_M** | 907 | 745/**1126**/1699 | N/A | 212 | N/A |
| 24.00 | Synthetic | **29_S_M** | 907 | 713/**1242**/1957 | N/A | 189 | N/A |
| 48.00 | Synthetic | **30_S_M** | 907 | 728/**1181**/1576 | N/A | 68 | N/A |
| 72.00 | Synthetic | **32_S_M** | 907 | 930/**1245**/1839 | N/A | 83 | N/A |

* Without EBSD data due to a broken ice thin section during sample cutting.



**Figure 2. Schematic drawings for Rig01 and Rig02 applied for ice grain-growth experiments are illustrated in (a) and (b), separately. (c) Temperature logging for ice grain-growth experiments conducted within Rig01. Red and purple arrows correspond to the termination of annealing experiments for synthetic ice samples and natural ice samples, respectively. The number beneath each**





**arrow marks the first number in the ice sample number, which is inherited in the ice sample number (Table 1). The colours of bag numbers correspond to different sample bins: blue represents bin01; green represents bin02. (d) Temperature logging for synthetic ice grain-growth experiments conducted within Rig01. Arrow marks the end of annealing experiments; sample numbers are marked beneath arrows.**

## 2.3 Electron backscatter diffraction (EBSD) data collection

Initial and annealed ice slabs were sectioned into ice slices of ~5mm using a band saw in a cold room at ~-20°C for microstructural analyses. For each synthetic ice slab with a thickness of ~15 mm and a diameter of 25.4 mm, one thin slice was sectioned. For each ice slab with a thickness of ~20 mm, i.e., synthetic ice with a diameter of 40 mm and natural ice, two thin slices (named with "A" and "B"; Table 1) were produced by sectioning at ~5 mm from the top surface and at ~5 mm from the bottom surface of ice slab (Fig. 1(b)). We did this to (1) maximize the number of 2D grains that can be captured for

statistical analyses, and (2) minimize the repeat count of the same 3D grain in different 2D sections, in the following microstructural analyses. For a few ice slabs, we only successfully subsampled one thin slice since the other slice was broken during bandsaw cutting (Table 1). We developed a fiducial marker system so that subsampled daughter thin slices can be easily reoriented to represent the original reference frame of their parent ice slabs. This is important since microstructural data collected from slices subsampled from the same ice slab will be combined for grain shape and crystallographic preferred

orientation statistics, which are sensitive to the imposed sample reference frame. The fiducial marker system involves:

(1) A fiducial line marked on the wall of ice slabs before sectioning (Fig. 1(b)).

(2) Fiducial marks on one of the surfaces of each daughter ice slice corresponding to the imposed top surface of the parent ice slab (Fig. 1(b)).

We prepared the surface of ice slices and collected cryo-EBSD data from ice surfaces following the procedures described by

Prior et al., (2015). The EBSD data were collected using a Zeiss Sigma VPFEGSEM combined with a Symmetry EBSD camera from Oxford Instruments. EBSD data were acquired at a stage temperature of~−95°C, with 2–5 Pa nitrogen gas pressure, 30 kV accelerating voltage, a beam current of~60 nA, and a step size of 30 µm. For Priestley ice samples, secondary electron (SE) images were collected simultaneously with EBSD data. We utilized the fiducial lines and marks (as mentioned in the last paragraph) to reorient ice slices subsampled from the same ice slab so that their sample reference frames remained consistent

during EBSD data collection.

During sample transportation and preparation for cryo-EBSD, microstructural modifications, such as grain growth, and changes in intragranular structure are likely negligible over the short timescales (within 30 minutes at $T$ < -20°C) involved here (Fan et al., 2022 and reference therein).



## 2.4 EBSD data processing

### 2.4.1 Phase segregation

For synthetic and Priestley (natural) ice samples, the EBSD map has each pixel attributed as "ice 1h" or "not indexed" during data collection. However, Priestley ice is bubble-rich (Thomas et al., 2021). Consequently, ice and air bubbles should be segregated for a more complete microstructural analysis.

Ice and air bubbles were segregated via thresholding on secondary electron (SE) images using MATLAB (Figs 3(a), 3(b)). SE
images show surface topography (e.g., Fig. 3(a)). The polished surface of ice has medium to dark grey colours. Bubbles are holes with light grey to white colours (e.g., Fig. 3(a)). The contrast between the flat surface and bubble holes was used to segregate ice and air bubbles.

Figures 3(b) and 3(c) illustrate the integration of the phase map from the SE image and EBSD pixel map for each Priestley ice sample. For each EBSD map, "not indexed" pixels that match "bubble" pixels (from the phase map) are attributed with a phase
of "bubble". This process ensures that scratches (example highlighted by green dashed ellipses in Figs. 3(a–b)), which were identified (wrongly) as bubbles due to similar grayscales during SE image thresholding, would not be mistakenly attributed as bubbles in EBSD maps.



*Natural, Priestley ice, starting material*





**Figure 3. Workflow for processing the microstructural data of ice samples. (a)–(d) represent the starting material of Priestley ice.**
**(a) Secondary electron (SE) image with the corresponding grayscale distribution. (b) Phase map after thresholding the SE image. Green dashed ellipse in (a) and (b) highlights a scratch that may be mistakenly identified as "bubbles" during thresholding. (c) Raw orientation map coloured by IPF-X, using the colour to indicate the crystallographic axes that are parallel to the x-axis. White pixels are not indexed. (d) Grain map after grain reconstruction with the input of integrated EBSD pixel data from (b) and (c). The bubbles are black. Pixels indexed as "ice 1h" are coloured by IPF-X. (e)–(f) represent the starting material of synthetic, medium-grained ice.**
**(e) Raw orientation map coloured by IPF-X (f) Grain map after grain reconstruction with pixels of "ice 1h" coloured by IPF-X. Ice grain boundaries are thin black lines.**

### 2.4.2 Grain reconstruction

We used the Voronoi decomposition algorithm in the MTEX toolbox (Bachmann et al., 2011) for grain reconstruction of raw EBSD data, for synthetic ice, and the integrated EBSD data (Sect. 2.4.1) for natural ice. Ice grains and bubbles were
reconstructed from pixels identified as "ice 1h" and "air bubbles", separately, with a boundary misorientation threshold of 10°. Firstly, we applied data filtering. Ice grains or bubbles containing less than 4 pixels were removed since they may result from mis-indexing. We also removed poorly constrained ice grains and bubbles with <50% indexed pixel coverage as well as ice grains and bubbles at the edge of EBSD maps. For synthetic ice, the filtered data were used directly for microstructural statistics (Figs. 3(e-f)). For natural ice, we applied interpolation on filtered EBSD data using the "*fill*" function in MTEX to optimize
the geometry of ice grains and air bubbles (e.g., Fig. 3(d)). The "*fill*" function populates less than 3% of the map area with interpolated pixels. After that, we reconstructed ice grains and bubbles using the interpolated data (e.g., Fig. 3(d)).

### 2.4.3 Microstructural parameters

We quantified the microstructure of ice with statistics of ice grain size and crystallographic preferred orientation (CPO). For each ice grain, grain size is defined as the diameter of a circle with its area equivalent to the ice grain area, i.e., area-equivalent
diameter. EBSD maps were used to generate ice CPO data with one point per pixel or one point per grain. To show CPO patterns more clearly, we contoured CPO data with a half-width of 7.5° based on multiples of a uniform distribution (MUD) of points. The CPO intensity was quantified by M-index (Skemer et al., 2005).

We quantified the microstructure of bubbles within Priestley ice using statistics of bubbles size, aspect ratio, and shape preferred orientation (SPO). Bubble sizes were calculated from area-equivalent diameters of bubbles, and it is the same method
as calculating ice grain sizes. We measured the aspect ratio, which is the ratio of the lengths of the long axis and short axis of an ellipse fitted to each bubble. Bubble shape preferred orientation (SPO) comprises the angles between a given vector (+x direction in this study; Fig. 3) and the bubble long axes estimated from the ellipse fit. The ellipse fit will provide an arbitrary long axis for bubbles with low aspect ratios, i.e., bubbles with a convex hull close to a circle. So only bubbles with aspect ratios higher than 1.5 were used for SPO analyses.



## 3. Results

Temperature as a function of time is shown in Figs. 2(c) and 2(d). Maps of ice microstructures, statistics of grain sizes, crystallographic preferred orientations (CPOs), and analyses of bubble sizes and shapes are shown in Figs. 4–6. Microstructural maps from selected areas are presented to illustrate microstructural features. Quantitative microstructural analyses are based on much larger areas (e.g., Fig. 3) than those presented in Figs. 4 and 5.

### 3.1 Temperature fluctuation

In Rig01 (Fig. 2(a), Sect. 2.2), the temperature within the silicon oil was generally maintained between -1 and 0ºC during a maximum experimental time of ~33 days (Fig. 2(c)). The temperature fluctuations within ice samples should be an order of magnitude less than in the silicon oil bath (Vaughan et al., 2017).

The temperature measured from reference ice samples in Rig02 remained stable at -0.25±0.25ºC during a maximum experimental time of ~5 days (Fig. 2(d)).

### 3.2 Synthetic, medium-grained ice

### 3.2.1 Ice microstructure

The starting material of synthetic ice exhibits a homogeneous microstructure with slightly irregular grain boundaries and very few (almost none) intragranular boundaries (Figs. 3(f), 4(a) and (b)). The distribution of grain sizes is skewed towards the left in the logarithmic scale, with a peak at ~1000 µm and a tail extending down to ~300 µm (Fig. 4(c)). The median and maximum ice grain sizes are 907 and 2409 µm, respectively (Table 1, (Fig. 4(c)).

After annealing experiments, the distributions of grain sizes remain skewed towards the left in the logarithmic scale, grain boundaries remain slightly irregular and the number of intragranular boundaries remains small—similar to the starting material (Figs. 4(a)–4(c)). With increasing annealing time, the number of ice grains within a given area decreases (Figs. 4(a) and (b)), consistent with increasing median grain size (Fig. 4(c)). The maximum grain size also increases with increasing annealing time, and it is generally 3 to 5 times of the median grain size (Fig. 4(c)).

### 3.2.2 Ice crystallographic preferred orientation

The starting material and samples annealed up to 194 hours, exhibit near-random *c*-axes orientations (Fig. 4(d)). Samples annealed to > 200 hours contain a small number of grains (<100; Table 1) and have *c*-axis CPOs characterized by multiple randomly distributed clusters (e.g., 5_S_M, 6_S_M; Fig. 4(d)) that are possibly a result of a small number of data sets (Monz et al., 2021).



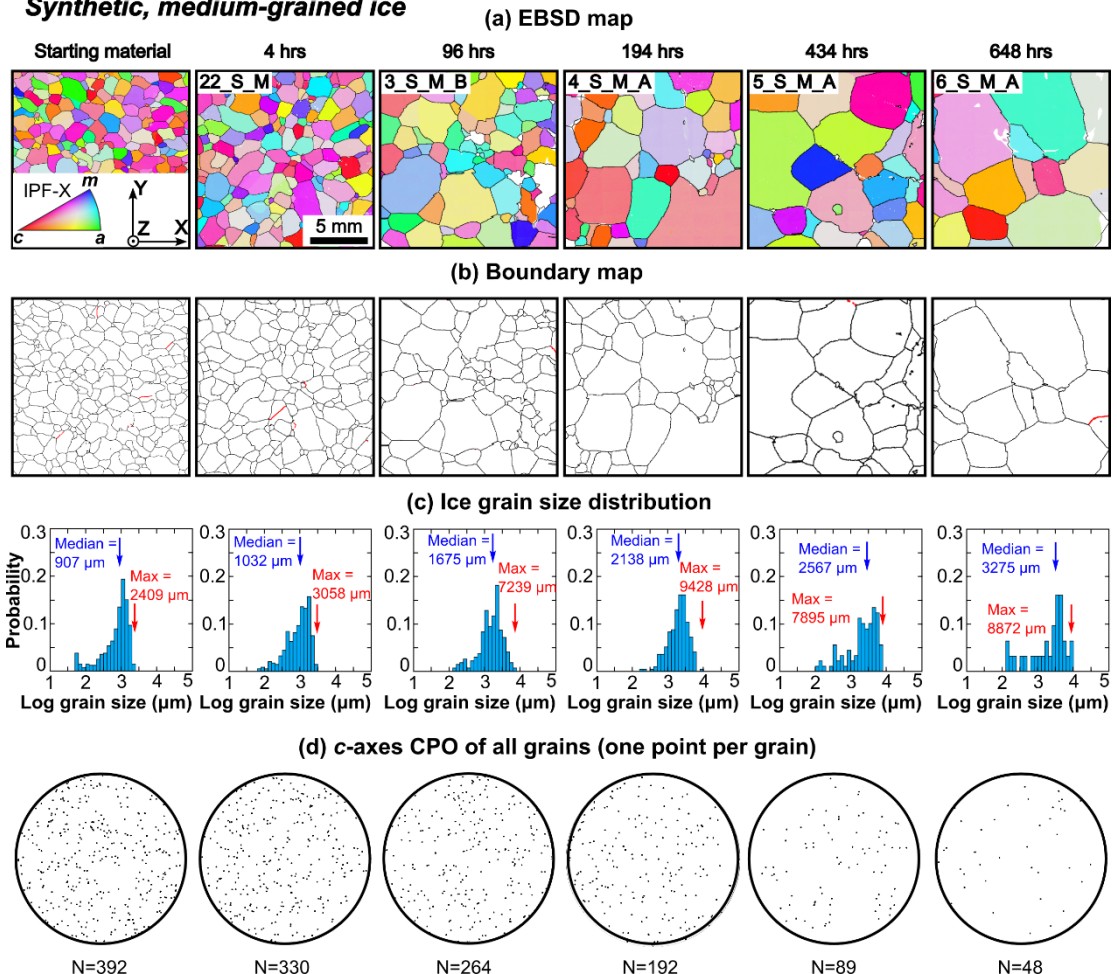

**Figure 4.** Microstructural and crystallographic preferred orientation (CPO) analyses of synthetic ice samples. For (a)–(d), columns from left to right represent samples with increasing annealing time. (a) Sub-areas of the ice orientation maps coloured by IPF-X. (b) Boundary maps. Black lines indicate ice grain boundaries; red lines indicate intragranular boundaries with misorientation angles of 4–10º. (c) Distribution of ice grain size presented in logarithmic scale. The blue arrow and red arrow indicate median and maximum ice grain sizes, respectively. (d) Preferred orientations of ice *c*-axes, which are displayed as point pole figures. The number of grains is given for each sample.

### 3.3 Natural, Priestley ice

#### 3.3.1 Ice microstructure

The starting material of Priestley ice is characterised by smaller ice grains with less irregular grain boundaries interlocking with larger ice grains with more irregular grain boundaries (Fig. 3(d) and 5(a)). Many ice grains are internally distorted, and have intragranular boundaries (Figs. 3(d), 5(a) and 5(b)). Grain sizes in the logarithmic scale are close to a normal distribution, with a peak at ~1000 µm (Fig. 5(c)). The median and maximum ice grain sizes are 917 and 6088 µm, respectively (Table 1, (Fig. 5(c)).





Ice microstructure does not change during annealing except for samples with ~72 hours (11_P_A) and ~800 hours (15_P_A) of annealing time, in which we observed individual ice grains that are much larger than their neighbouring ice grains; very large grains are yellow and grey, while most of the other grains are red in crystallographic orientation maps (Fig. 5(a)). For these two samples (i.e., 11_P, 15_P), the maximum ice grain size is ~12–18 times of median ice grain size (Fig. 5(c)).

**3.3.2 Bubbles**

For all samples, bubbles occur both on ice grain boundaries and within ice grains (Figs. 3(d), 5(a) and 5(b)). For the starting material, bubble-size distribution is skewed toward the right in the linear scale, with a peak at ~200 µm and a tail extending up to ~1000 µm (Fig. 5(d)). The median bubble size and bubble-size distribution remain similar for samples annealed to less than ~360 hours (Fig. 5(d)). For samples annealed to ~360 and ~800 hours (i.e., 13_P, 15_P), there is a slight increase in the

275 median bubble size resulting from a subtle decrease in the proportion of small bubbles (< 500 µm) and a subtle increase in the proportion of large bubbles (> 500 µm) (Fig. 5(d)). Abnormally large bubbles are not observed in the microstructural maps, and the maximum bubble size is 4 to 5 times the median bubble size for all the samples (Figs. 5(a), (b) and(d)). For the starting material, shape preferred orientation (SPO) of bubbles is characterised by a primary maximum at ~30º to the flow direction (x-axis) and a secondary maximum at ~120º to the flow direction (Figs. 5(e) – (f)), similar to patterns observed in a wide range

of Priestley ice core samples (Thomas et al., 2021). For samples annealed to <100 hours, the bubble-number frequency of the primary SPO maximum decreases and the bubble-number frequency of the secondary SPO maximum increases with increasing annealing time (Figs. 5(e) – (f)). After ~100 hours, the bubble SPO is close to isotropic (Figs. 5(e) – 5(f)).





**Natural, Priestley ice**

**(a) EBSD map**

Starting material | 4 hrs | 72 hrs | 174 hrs | 360 hrs | 780 hrs

**(b) Boundary map**

**(c) Ice grain size distribution**

**(d) Bubble size distribution**

**(e) Bubble SPO (for bubbles with aspect ratio > 1.5)**

**(f) Bubble map with each bubble coloured by its long-axis orientation (aspect ratio > 1.5)**





**Figure 5. Microstructural analyses of Priestley ice samples. For (a)–(d), columns from left to right represent samples with an increasing annealing time. (a) Sub-areas of the ice orientation maps coloured by IPF-X. (b) Boundary maps. Black lines indicate ice grain boundaries; red lines indicate intragranular boundaries with misorientation angles of 4–10º. Black blobs indicate bubbles. (c) Distribution of ice grain size presented in logarithm scale. The blue arrow and red arrow indicate median and maximum ice grain size, respectively. (d) Distribution of bubble size. The green and gold arrows indicate median and maximum bubble size, respectively. (e) Shape preferred orientation (SPO) of bubbles with aspect ratios larger than 1.5. (f) Three examples illustrate the SPO of individual bubbles. Each bubble is coloured by the angle between the bubble's long axis and the +x axis. The black line within each bubble represents the orientation of the bubble's long axis. For each sample, SPO for bubbles with aspect ratios higher than 1.5 is presented at the bottom left.**

### 3.3.3 Ice crystallographic preferred orientation

Patterns of ice *c*-axes are generally characterised by two clusters (Fig. 6(a)). The primary *c*-axes cluster is sub-perpendicular to the ice flow (shear) direction (Thomas et al., 2021); the secondary *c*-axes cluster is at 40-50º to the primary cluster. Grains with abnormally large sizes (> 8000 μm; Sect. 3.3.1) have *c*-axes within the secondary *c*-axes cluster (star marks in Fig. 6(a)). Figure 6(b) summarises the M-index as a function of annealing time. The M-indices of samples with annealing time up to 380 hours fall within the range of M-indices of un-annealed Priestley ice samples that were collected from > 10 m depth (Thomas et al., 2021). The M-index of sample with 780 hours of annealing is lower than the other annealed samples, approaching the M-index observed in the upper 10 m of the Priestley ice core (Thomas et al., 2021).

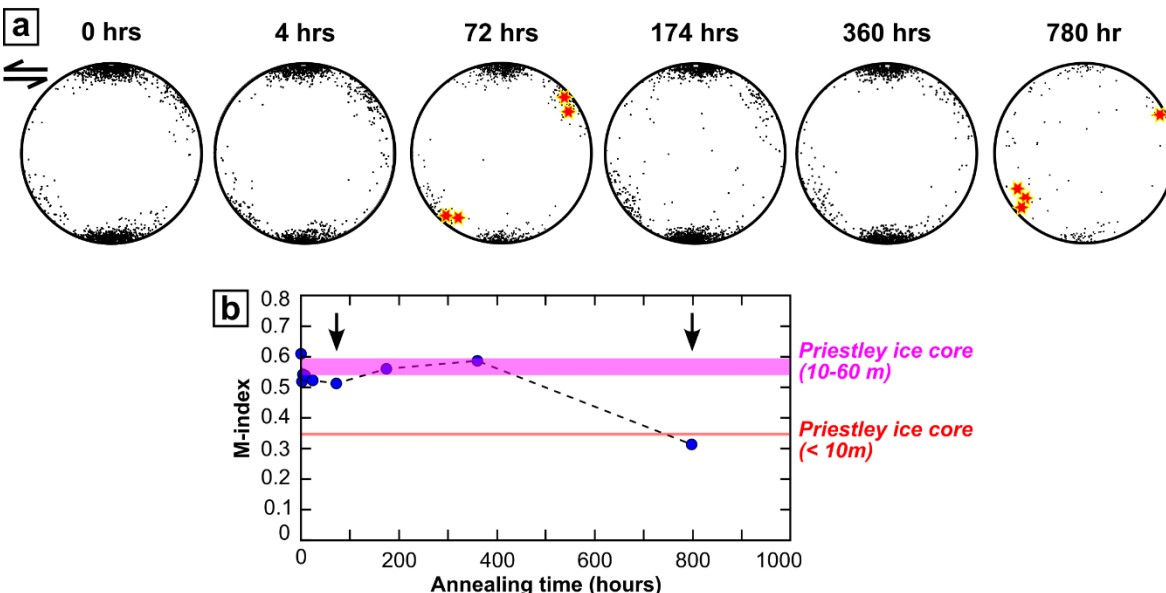

**Figure 6. CPO analyses of Priestley ice samples. (a) Preferred orientations of ice *c*-axes, which are displayed as point pole figures with one point per grain. The *c*-axes orientations of abnormally large grains (grain size > 8000 μm) are highlighted by star marks (columns 3 and 6). (b) M-index, which was calculated from one point per grain, as a function of annealing time. Vertical black arrows indicate samples with abnormal grain growth. Horizontal lines mark the M-indices (with one point per grain) of ice samples at different depths of the Priestley ice cores (pink: 10-60 m; red: < 10 m) as reported by Thomas and others (2021).**




## 4 Discussion

### 4.1 Inferences from ice grain size evolution

#### 4.1.1 Synthetic pure water ice

The median grain size of the synthetic ice increases with time and the rate of growth decreases as grain size increases (Fig. 7). Furthermore, the shape of the grain size distribution does not change significantly during annealing (Fig. 5(c)). These observations are characteristics associated with "normal grain growth". The average grain size of single-phase polycrystalline materials, such as olivine, quartz, and ice, during normal grain growth can be described in a power-law form:

$$D^n - D_0^n = kt, \tag{1}$$

where $D$ is the average grain size at time $t$; $D_0$ is the average grain size at the starting of annealing; $n$ is the grain size exponent; $k$ is a thermally activated rate constant (Evans et al., 2001; Alley et al., 1986; Karato, 1989; Covey-Crump, 1997).

We applied the method from Bons and others (2001) and Azuma and others (2012) to quantify grain growth parameters. The average grain size at the starting of annealing, $D_0$, is given as:

$$D_0^n = kt_0, \tag{2}$$

where $t_0$ is an unknown hypothetical incubation time needed to reach the average grain size at the starting point of grain growth. Combine Eq. (1) and Eq. (2), and then the grain size change during the normal grain growth of polycrystals can be expressed as:

$$D^n = k(t + t_0). \tag{3}$$

Take the logarithm of Eq. (3):

$$n\log(D) = \log(k) + \log(t + t_0). \tag{4}$$

Equation (4) can be transformed to:

$$\log(D) = \frac{1}{n}\log(t + t_0) + \frac{1}{n}\log(k). \tag{5}$$

Equation (5) contains three interdependent unknown parameters, including $n$, $k$, and $t_0$. We applied numerical iteration to Eq. (5) using median grain size and annealing time as inputs. Figure 7(a) shows the growth exponent, $n$, of 2–3 is required to
achieve a relatively robust fit to the measured grain size change, i.e., with the value of $R^2$ (measurement of the goodness of fit) greater than 0.95.







**Figure 7. (a) Summary of ice median grain size evolution with an increasing annealing time for synthetic ice, and Priestley ice. (b) Comparing grain-growth curves with different values of _k_. Measured ice grain size is shown as interquartile range (IQR); the solid circle is the median grain size; grey whiskers represent 50% of the data.**

### 4.1.2 Natural, Priestley ice

For Priestley ice samples, the median ice grain size stays within the range of values reported for different sub-samples of the Priestley ice core (Thomas et al., 2021), suggesting that insignificant normal grain growth has occurred during annealing (~33 days in this case).

Grain sizes changes are too small to evaluate the grain-growth parameters (i.e., grain-growth exponent $n$, rate constant, $k$) of natural ice. To quantify maximum grain growth rates in natural ice, we calculated what parameters ($n$ and $k$) would give grain growth too small for us to measure. First, we fixed $n$ to a value of 2, a value for perfect normal grain growth (Covey-Crump, 1997) that is constrained for bubble free synthetic ice (Azuma et al., 2012), and it gives a good fit to our synthetic data. This allows us to calculate a limiting value for $k$, representing the grain boundary mobility during normal grain growth. Figure 7(b) compares the modelling result with measurements, and it shows that to reproduce the grain size change in natural ice, the $k$ value needs to be at least two magnitudes lower than the synthetic ice (compare red dots with blue and green curves). This





observation suggests, if the grain growth process in natural ice and synthetic ice is the same, i.e., dominated by normal grain growth, then a reduced grain boundary mobility and driving force (as indicated by estimated low value of $k$), should govern grain size change in natural ice. Second, we fixed the value of $k$ ($k = 4.6\times10^{-6}$ mm$^2$s$^{-1}$) measured from synthetic ice (Sect.

4.1.1) while varying the value of $n$. To reproduce the grain size change in natural ice, the $n$ value needs to be ~50. This observation suggests if the grain boundary mobility and driving force are the same between synthetic and natural ice, then processes that are different from normal grain growth (as indicated by estimated high value of $n$) should govern grain size change in natural ice. We suggest, the grain-growth inhibition in natural ice is controlled by both reduction of grain boundary mobility and driving force (as discussed in Sect. 4.2) and processes that are different from normal grain growth (as discussed

in Sect. 4.3).

### 4.2 Evaluating the role of bubbles on grain-growth inhibition in Priestley ice

During ~400 hours of annealing, the bubble size remains unchanged; however, after ~800 hours of annealing, the number frequency of small bubbles (< 500 μm) decreases whilst the number frequency of large bubbles (> 500 μm) increases (Fig. 5(d); Sect. 3.3.2). These observations suggest a slow growth of air bubbles (Alley et al., 1986; Shewmon, 1964). Bubble shape

preferred orientation (SPO) was modified during annealing: the primary SPO maximum weakens with increasing time in the first ~100 hours; after ~100 hours, the SPO becomes isotropic (Figs. 5(e), 5(f); Sect. 3.3.2). This observation is consistent with previous studies suggesting bubbles tend to return to a spherical shape via vapour diffusion driven by curvature and surface tension (Hudleston, 1977; Alley and Fitzpatrick, 1999).

The grain-growth rate of Priestley ice is much slower than synthetic ice during annealing experiments (Fig. 7; Sect. 4.1). The

migration rate of a grain boundary is a function of grain boundary mobility and driving force (Humphreys et al., 2017). Previous studies on metal, rock-forming minerals, and ice show, the driving force and kinetics of grain boundary migration are influenced by microstructures (Humphreys et al., 2017; Herwegh et al., 2011; Fan et al., 2021c; Azuma et al., 2012; Kilian et al., 2011). For example, secondary phases can introduce a dragging force and thus reduce the driving force for grain boundary migration (Herwegh et al., 2011; Humphreys et al., 2017; Kilian et al., 2011). Neighbouring grains with different orientations

may exhibit a contrast of dislocation densities during deformation (Vaughan et al., 2017; Fan et al., 2021c; Boneh et al., 2017). The dislocation-density contrast can drive the migration of grain boundaries from grains containing low dislocation densities towards grains with high dislocation densities (Hirth and Tullis, 1992). We compared key microstructural differences between the starting materials of synthetic ice and Priestley ice to address possible microstructural controls on the inhibition of grain growth in Priestley ice; these microstructural differences include:

(1) Bubble content. The Priestley ice contains many more bubbles than the synthetic ice as revealed by the SE image data (compare Fig. 4(a) and Fig. 5(a).

     (2) Insoluble and soluble impurities. Natural ice contains insoluble impurities, such as $CaSiO_3$ and $SiO_2$, and soluble impurities, such as ions produced from dissolved salts (Gow, 1968; Gow and Williamson, 1971; Svensson et al., 2005; Faria et al., 2010; Baker et al., 2003; Weikusat et al., 2017b). Preliminary work (Rilee Thomas, personal communication) shows that



Priestley ice contains dissolved impurities and sub-micrometre to millimetre particles. The methods applied in this study do not enable us to locate impurities within grains and/or at grain boundaries. In contrast, the synthetic ice was produced from ultra-pure water (Sect. 2.1); therefore, the content of insoluble and soluble impurities should be negligible.

(3) Geometrically necessary dislocation (GND) density. The Priestley ice develops intragranular boundaries (Fig. 5(b); Sect. 3.3.1), indicating a relatively high GND density. In contrast, the synthetic ice has few intragranular boundaries (Fig.

4(b); Sect. 3.2.1), indicating a relatively low GND density.

(4) CPO. The Priestley ice has a strong CPO whilst the synthetic has a CPO that is close to random (compare Fig. 6(a) with Fig. 4(d)).

Evaluating the impact of insoluble and soluble impurities on the normal grain growth of ice would require additional data input, most particularly the distribution of impurities relative to the microstructure. We have data on CPO, and we can use

intragranular distortion as a proxy for GNDs. However, exploring their effects requires an extensive modelling programme that is beyond the scope of this paper. Thus, in the following paragraphs, we will focus on evaluating the impact of bubbles on the inhibition of ice grain growth.

Bubble density, i.e., bubble number per unit area, is one of the key parameters that can help us to understand how bubbles interact with grain boundaries (Duval, 1985; Durand et al., 2006; Azuma et al., 2012). We separately calculated the density of

bubbles at ice grain boundaries and bubbles in the interior of ice grains (Fig. 8(a)). We also calculated the density of bubbles with different size ranges. We use the measured bubble density relative to the starting material to better visualise the relative change of bubble density during annealing. Up to ~400 hours, the bubble density remains similar for all bubbles (black circles, Fig. 8(b)), suggesting an insignificant sample-to-sample variation and no significant changes in bubble number.



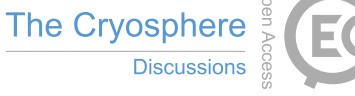

**Figure 8. (a)** An example that illustrates the discrimination of bubbles on ice grain boundaries from bubbles in grain interior. **(b)** The evolution of normalised density with an increasing annealing time for all bubbles, bubbles on grain boundaries, and bubbles in grain interior. **(c)** The evolution of normalised density with an increasing annealing time for bubbles with different sizes that are on grain boundaries or in grain interior.

The bubble density remains stable before ~400 hours of annealing for both bubbles on grain boundaries and bubbles in grain interiors (triangle and square marks, Figs. 8(b–d)). This observation suggests that grain boundaries do not move relative to bubbles, probably due to a strong pinning effect of bubbles and/or other secondary phases along grain boundaries in the starting material. At ~800 hours of annealing, the density of bubbles in grain interior increases, whilst the density of bubbles on grain boundaries decreases (triangle and square marks, Figs. 8(b–d)). This observation suggests that some grain boundaries have moved from their original position, whilst bubbles that were along these boundaries have not moved. Consequently, for these





grain boundaries, the bubble-pinning force decreases, and the boundary migration rate is enhanced. The contrast of the density of bubbles at grain boundaries between the early (up to ~400 hours) and late part of our annealing experiments indicates that the ice-grain growth rate should be different. Previous modelling results show, during grain growth, the value of rate constant, $k$, which is a function of grain boundary mobility and driving force (Evans et al., 2001), varies before the ice-bubble matrix reaches an equilibrated state—i.e., the contrast between the length-scales of bubble-spacing and ice grain size becomes

relatively constant (Roessiger et al., 2014). Consequently, the grain-growth rates of natural ice at different stages, i.e., before and after reaching an equilibrated microstructure, might be different (Roessiger et al., 2014). We speculate that the bubble-ice aggregate in Priestley ice was far from reaching an equilibrated microstructure within a relatively short amount of time (~33 days in this study). Thus, for the Priestley ice, the value of rate constant, $k$, between short (e.g., several months) and long (e.g., thousands of years) annealing times might be different. Further studies should introduce numerical models (e.g., Elle, Jessell

et al., 2001), to assess parameters at different stages of grain growth of natural ice.

Previous studies suggest grain boundary can modify the surface tension of air bubbles; consequently, bubbles can be dragged by grain boundaries and move via water-molecule transportation (Azuma et al., 2012). We did not directly observe the movement of air bubbles in this study. This is because our data are "snapshots" of ice and bubble microstructures. Understanding the kinematics of bubble movement requires additional input from in-situ annealing experiments, where bubble

positions can be continuously monitored.

**4.3 Abnormal grain growth in Priestley ice**

Some of the annealed Priestley ice samples (11_P, 15_P) have grains that are much (~12–18 times) larger than the average grain size (Figs. 5(a), 5(c); Sect. 3.3.1). Abnormally large grains are not observed in the Priestley ice cores (Thomas et al., 2021), and we suggest that these abnormally large grains were produced during annealing.

Previous experiments on metals, ceramics, and rock-forming minerals attribute the production of abnormally large grains within a matrix that initially contains much finer grains to the operation of rapid preferred grain growth, i.e., abnormal grain growth (Hillert, 1965; Boneh et al., 2017; Cooper and Kohlstedt, 1984; Bae and Baik, 2005; Dillon et al., 2010). For non-film materials, the occurrence of abnormal grain growth usually correlates with initially slow grain growth and some grains being surrounded by boundaries with different characteristics to the bulk (Hillert, 1965; Gladman, 1966; Rollett and Mullins, 1997;

Humphreys et al., 2017). The Priestley ice samples were initially bubble-rich, and we infer that the grain growth of the ice matrix was strongly inhibited by bubble-pinning (Sect. 4.3).

For annealed Priestley ice samples, most grains that are not abnormally large have $c$-axes within the primary $c$-axes cluster (Figs. 6(a), 6(b); Sect. 3.3.3). On the contrary, all the abnormally large grains have $c$-axes within the secondary $c$-axes cluster (Figs. 6(a), 6(b); Sect. 3.3.3). We segregated grains with $c$-axes at 0–15° to the centre of the primary maxima from grains with

$c$-axes at 0–15° to the centre of the secondary maxima to assess the dislocation-density contrast between grains with different orientations (Fig. 9(a)). For each selected grain, we calculated the length density, i.e., length per unit grain area, of intragranular boundaries with misorientation angles higher than 4°. Intragranular boundaries with low misorientation angles (< 4°) are subject





to large angular error and therefore not considered (Prior, 1999). The boundary length density in grains within the primary *c*-axes cluster is generally higher than that in grains within the secondary *c*-axes cluster; such contrast is especially obvious

within samples that develop abnormally large grains (11_P, 15_P; indicated by vertical arrows) (Fig. 9(b)). The dislocation-density contrast is considered as an important driving force for grain boundary migration from areas with lower dislocation densities towards areas with higher dislocation densities (Hirth and Tullis, 1992; Humphreys et al., 2017). Consequently, we infer that the migration rate for boundaries between grains with *c*-axes at primary and secondary clusters is faster than the migration rate for grain boundaries between grains with *c*-axes within a single cluster.

Discussions in the last two paragraphs suggest, the abnormal grain growth observed in some of the annealed Priestley ice samples should be tightly correlated to (1) bubble pinning, which inhibited the movement of the boundary matrix, and (2) dislocation-density contrast, which favours the selected growth of grains with low dislocation densities at the cost of neighbouring grains with high dislocation densities (Fig. 10). This is consistent with previous theoretical predictions (Hillert, 1965; Gladman, 1966; Rollett and Mullins, 1997; Humphreys et al., 2017), experimental observations (Boneh et al., 2017;

Cooper and Kohlstedt, 1984; Bae and Baik, 2005), and numerical modelling results (Doherty et al., 1997; Srolovitz et al., 1985).

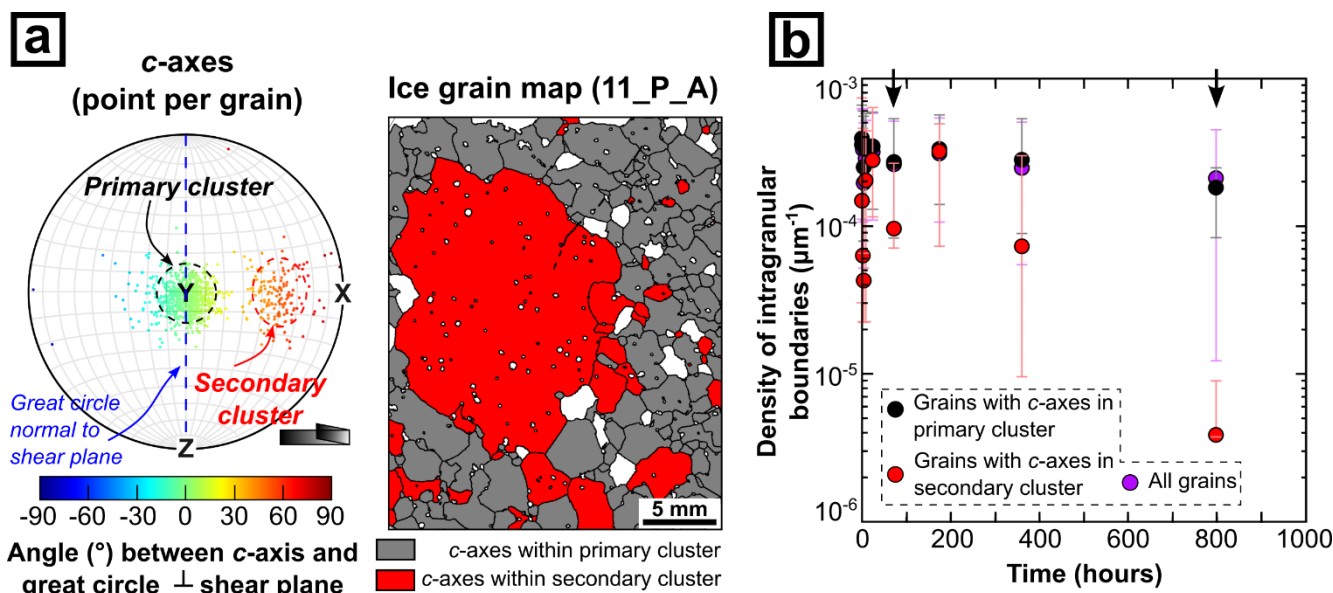

**Figure 9. Microstructural statistics for grains with different orientations. (a) Illustrating the segregation between grains with *c*-axes within the primary *c*-axes cluster, and grains *c*-axes within the secondary *c*-axes cluster. (b) Comparing the length density, i.e., length**
**per unit area, of intragranular boundaries (misorientation angle > 4°) amongst grains with different orientations.**

Samples (12_P, 15_P) that exhibit abnormal grain growth have M-indices that are generally lower than the other samples, and they are close to the M-indices of the upper 10 metres of unannealed Priestley ice core (Sect. 3.3.3). Temperature profile of the borehole shows that the summer ice temperature increases drastically, up to ~0°C in the top 10 metres of the Priestley glacier (Thomas et al., 2021). Consequently, annual thermal annealing should be a norm in the shallow part of Priestley ice. A



similar M-index between annealed sample with abnormal grain growth and unannealed shallow (< 10 m depth) Priestley ice core suggests the abnormal grain growth might be active at the shallow part of Priestley glacier. Annealing experiment of ice sample deformed by direct shear also shows an enhancement of the secondary *c*-axes cluster (Journaux et al., 2019), although there is no evidence of the grains within the secondary *c*-axes cluster being substantially larger than other grains after annealing.

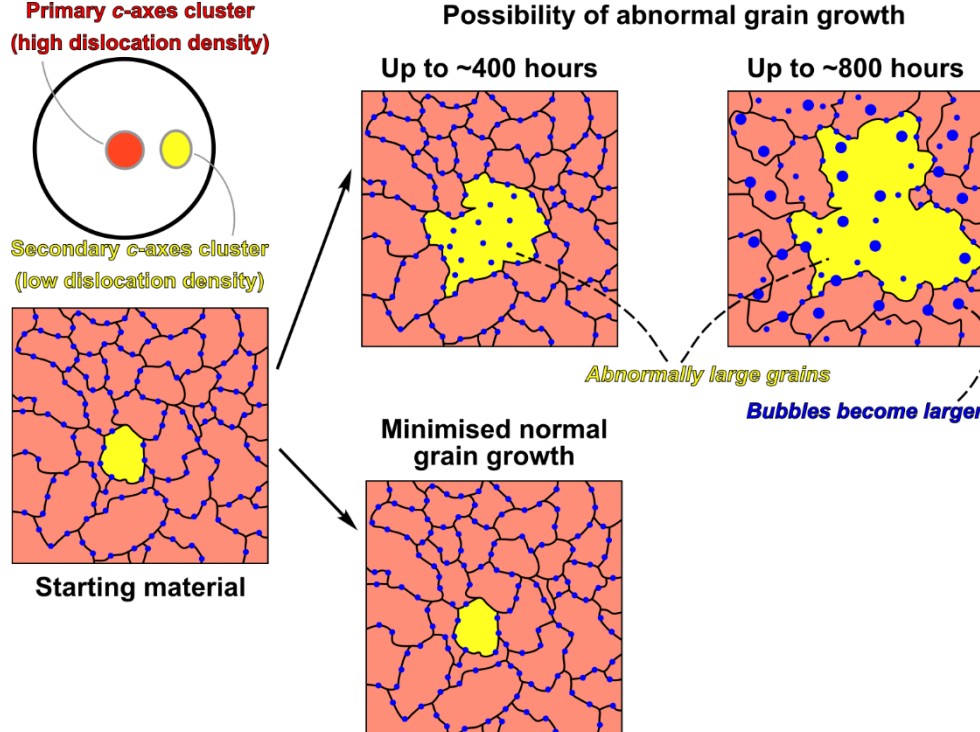

**Figure 10**. Schematic drawing of microstructural development of Priestley ice during annealing.

## 5 Conclusions

1. Annealing experiments at 0℃ were conducted on synthetic, ultra-pure water ice samples, and natural, Priestley ice samples. The grain size of synthetic samples increases with annealing time, such grain-size evolution can be explained by normal grain growth. On the contrary, the grain size of natural ice sample barely changes within a month, and it cannot be simply explained

by normal grain growth. This observation also indicates that a relatively short-period of abrupt temperature change during sampling of natural ice cores should have an insignificant impact on average ice grain size.

2. The inhibition of grain growth in natural ice is correlated with the observation of (1) bubbles at ice grain boundaries, and (2) development of abnormally large grains that do not exist in the starting material. Bubble pinning reduces the driving force for grain boundaries migration; abnormal grain growth introduces an additional grain-growth process to normal grain growth.

Together, bubble pinning and abnormal grain growth govern the grain size change in natural ice.

3. The density of bubbles at grain boundaries in natural ice changes during annealing. This observation suggests the driving force and kinetics of grain growth in natural ice, which is influenced by bubble pinning, should also vary during annealing. Consequently, we speculate that grain growth in natural ice might comprise more than one stage and it should correspond to more than one set of grain-growth parameters.

4. Abnormally large grains contain low dislocation density, whilst their neighbouring grains contain high dislocation density. This observation suggests dislocation-density contrast can provide driving force for abnormal grain growth. Moreover, widespread bubble pinning inhibits the grain boundary migration of the overall ice matrix. Bubble pinning and intergranular dislocation-density contrast are tightly correlated with abnormal grain growth.

5. Annealed Priestley ice samples that contain abnormally large grains exhibit a weaker CPO intensity compared with other
annealed samples without abnormal grain growth. This observation is consistent with the CPO of the shallow part of the Priestley ice core, which experienced thermal annealing due to annual heating, being weaker than the deeper part of Priestley ice core. These observations suggest abnormal grain growth might be a norm at the shallow part of Priestley glacier.

*Data availability.* Data will be available via Mendeley Data (open-access data share run by Elsevier with a permanent doi)
once the manuscript is accepted.

*Author contributions.* Conceptualisation: SF and DJP. Methodology: SF and DJP. Resources: SF, DJP, BP, HB, and LD. Investigation: SF. Data curation: SF. Software: SF. Formal analysis: SF. Validation: SF, DJP, and CQ. Writing (original draft): SF. Writing (review & editing): SF, DJP, SP, QC, DLG, and TFH.


*Acknowledgements.* We are thankful to Julie Clark for providing the cold room facility at University of Otago. SF was supported by a research grant from New Zealand Ministry of Business, Innovation and Employment through the Antarctic Science Platform (ANTA1801) (grant no. ASP-023-03), and a New Zealand Antarctic Research Institute (NZARI) Early Career Researcher Seed Grant (grant no. NZARI 2020-1-5). DJP was supported by two Marsden Funds of the Royal Society
of New Zealand (UOO1116 and UOO052) to DJP. CQ was supported by a NSFC grant (41972232). DLG was supported by a NASA fund (NNX15AM69G).

*Competing interests.* The authors declare that they have no conflict of interest.

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
