# Peer review of "Grain growth of natural and synthetic ice at 0°C"

_The Cryosphere, 2022_

## Referee Comment (RC1)

Referee comment to Fan et al. "Grain growth of natural and synthetic ice at 0°C"

Fan et al. provide an interesting and well-structured manuscript regarding the differences in grain growth of natural and synthetic ice. Despite being investigated for several decades, grain growth of ice crystals remains a topic with many open questions. This is mainly due to the challenges in observing grain growth in situ, i.e. in glaciers or ice sheets, and in setting-up laboratory experiments with boundary conditions similar to natural conditions (time, strain). The presented manuscript compares synthetic ice samples with natural samples from Antarctica and discusses the differences in the microstructure observed during the experimental time frame.

The manuscript is well-written, provides good figures, and the experiments have been carried out thoroughly. The topic is of interest for the community and the data gathering and analysis is well done as far as I can evaluate it. However, the motivation, and especially the decision to conduct experiments at 0°C, should be explained in more detail. I am not an expert in annealing studies and 0°C are usually avoided at all costs in microstructural ice core research. With some work on the issues mentioned below, I am sure that the manuscript can be published after minor revisions in *The Cryosphere*.

**General comments:**

1. I agree that crystal grain growth in ice is far from understood and that temperature certainly plays a role. However, the motivation of the manuscript remains a little unclear to me. You present examples (l.72-78), but these are merely very specific cases. Mechanical drilling is preferred over hot-water drilling for the majority of ice cores, especially for deep ice cores, and subglacial water inflow into a borehole is scarce and does not impact the already drilled ice, thus a maximum of a few meters. The vast majority of natural ice samples are well preserved for microstructure analysis or are discarded right away. Hence, I would encourage you to elaborate a bit more on the perks of the new insights regarding grain growth at 0°C (e.g. maybe that hot-water drilling could be used more frequently)?

2. The language is overall fine, but more (small) mistakes occur towards the end of the manuscript. I try to mention them below, but another spelling check would be beneficial.

3. I would be interested in a real photograph of the annealing experiment. Fig. 2 gives a good overview, but a photograph in the Appendix would be useful.

**Specific comments:**

l. 22: natural ice is characterized
l. 23: I think you mean "grain boundaries are pinned by bubbles"
l. 26: in the grain interior
l. 27: It is not totally clear what you mean by one stage, maybe add half a sentence to describe it.
l. 30: the wording is unprecise, "ice grain growth" or "growing of the ice crystals" might be better
l. 31: I would like to read one last sentence in the abstract briefly describing what your results mean
l. 44: delete "statistics"
l. 56: add "comparably impurity-rich"; ice is still a very pure material.
l. 57: Weikusat et al., 2017b deals only very briefly with impurities and does not investigate them; for more recent work refer to e.g. Bohleber et al., 2023, Stoll et al., 2022, 2023.
l. 62: For recent microstructural results from a deep ice core maybe add Stoll et al., 2021b.
l. 64: The sentence is certainly true, but it is important to add that several processes are not understood yet. Especially the relationship between grain growth, fabric/deformation, and impurities/bubbles remains unclear. Zener pinning has, to my knowledge, only been observed very rarely in ice and the presented evidence raises some doubts. You could cite Stoll et al., 2021a and references within for a recent review and overview of the topic without going into to details here (since it is not the scope of this manuscript).

l. 67: maybe replace original with host

l. 74: The entire ice core or "just" the outer rim? As mentioned above, hot-water ice cores are not primarily used for microstructural analysis as far as I know.

l. 76 ff: Backflow of water is luckily not usual, but occurs in special occasions, such as the EPICA drilling, and is tried to be avoided. Furthermore, there are many problems if such backflow and refreezing occurs, the impacted ice will certainly not be used for "normal" microstructural analysis. Mentioning this rare occasion as a major objective is thus not very strong. Similarly, natural samples are usually well preserved or discarded before analysis.

Rephrase it to "can occur" to avoid drawing a misleading image here and rethink the justification of your objectives, I am sure there are better fitting options to address.

l. 83: Merge both sentences: ..kinetics to better understand grain growth in natural ice".

l. 92: space between of25.4

l. 96: replace eliminated with discard/vanish/exude or something similar

l. 98: ice slabs are used

l. 99: thickness of ice slabs

l. 100: is there a more specific name than Antarctic ice core no 30.?

l. 104: Fig. number missing

Fig. 1: Change to Thickness. I think it would be helpful to clearly define your natural ice samples once and then just refer to it, thus either natural or Priestley ice (I would suggest the latter).

l. 110: Cold laboratory

l. 115: the bins

l. 118: hamper/impact

l. 124: There might be more precise words than recharging and maintained. Once can be deleted.

l. 142: Maybe add dewar flask/bottle for readers not acquainted with laboratory terms.

Table 1: Clarify the parameter for grain size, is it diameter?

I am confused by sample 12 and 13, the ice grain numbers don´t add up as they do in the other samples. If they are not available it´s fine, but in sample 15 N/A does not have an impact on the total number. Please clarify.

*You usually refer to a "thin slice", here to a "thin section" – stick with one term or do they differ?

Fig. 2b): The small plot is a bit confusing. Maybe you can think of a clearer way to present it, if this information is crucial.

l. 155: space at 5mm

l. 169: Describe briefly how you prepared the slices, did you microtome the surface?

Fig. 3: Put the scale in panel a and e.

l. 213: CPO already introduced, the same applies for SPO later in the text.

l. 217: Briefly introduce the M-index and add "the".

l. 219: …bubbles, as done for calculating grain sizes.

l. 222: delete "will"

l. 226-229: Is this needed? You explain the different aspects in more detail, I think you don´t need this introduction.

l. 231-235: I think it´s important to have a temperature record to evaluate the experiments, but a designated subsection might not be needed. Maybe you find a way to implement it in the other sections; if not leave it like it is.

Section 3.2: medium is a relative term (as long as you don´t define it), I am not sure if the grain size has to be in the title

l. 249: delete to. I think the use of "c-axis CPOs" is redundant, CPOs usually refer to the c-axis if not stated otherwise. You don´t show a-axes data here, so rephrase to e.g. "…and c-axis patterns are characterized by …

Fig. 4c): The x-axis could be designed more accessible with e.g. 10^x

l. 261: small ice grains, otherwise you need something to compare it with

l. 268: I suggest to clearly state the crystal orientations instead of the colours, which is confusing to readers not used to CPOs and the colour legend

l. 269: 12-18 times larger than that of/ 12-18 times the value of the median ice grain size

276: Delete the part about abnormal bubbles if you don´t discuss it later

l. 277: space and(d)

l. 278: SPO introduced already

l. 279: the last part belongs into the discussion section

l. 281: increases with time

Fig.5c) The x-axis could be designed more accessible with e.g. 10^x

Is there any meaning to the black-white bars below c and e? The caption could be shortened by referring to Fig. 4.

l. 294: CPO patterns are generally…

Fig. 6 caption: delete point, only pole figure. Thomas et al. (2021), similar with Bons and others and Azuma and others -> consistency in referencing

Fig. 7 Annealing time in s is tricky, try to stick to time formats already used (hours preferably). The same goes for grain size, so far you used microns. Solid circles are

l. 344: Fig. 7 does not compare, it displays the data which is a comparison. Rephrase.

l. 347-355: Challenging to read as you refer to sections not read yet. Maybe rephrase or switch order.

l. 367: by the microstructure

l. 372: We compare the following microstructural differences…:
Delete "these microstructural differences include"

l. 375: Can you quantify "many more" with your data?

l. 379: Weikusat et al., 2017b deals only very briefly with impurities and does not investigate them; for more recent work refer to e.g. Bohleber et al., 2023, Stoll et al., 2022, 2023. For readability it would be good to stick with soluble or dissolved impurities.

The discussion of impurities in your samples is tricky, since no data is published (yet). All natural ice contains impurities, this is thus not a necessary statement. I think you can shoren this paragraph and mention that impurtiies play a role (even though the details are unclear, see Stoll et al., 2021a), but you can´t discuss this point due to missing data (which is totally fine, investigating impurities would be another major approach). This is partially done in later sentences, could be easy to combine there

l. 386: strong CPO pattern – which one?

l. 395: To avoid confusion, clearly define that you are talking about the number to area ratio as density, and did not measure the density of the bubble.

l. 397: bubble density remains similar for all bubbles? Clarify that you talk about different (location) type of bubbles and maybe name them instead of using "all bubbles"

Fig. 8: You use bubble density in the text, but not on the respective y-axis. Explain it in the text/caption and then use it accordingly.

l. 405: To me, the density does not remain stable in 8d) for the first 400 hours.

l. 407: density of bubbles evokes material density thoughts, stick to bubble density

l. 410: long complicated sentences, try to break it up

l. 421: give examples of previous studies or use the singular. Grain boundaries can modify.

l. 422: Simply say that the applied methods cannot investigate bubble movement, that´s totally fine and does not have to be defended

l. 427: delete much and directly name the size difference

l. 430: replace production with "development" or "attribute abnormal grain growth with" -> the sentence can be shortened

l. 433: What is non-film material?

l. 434: This is confusing. Grains are not really surrounded by grain boundaries, but contain planar defects/dislocations separating the material inhibiting different orientations. Give examples of these characteristics. Grain growth of the ice matrix is redundant and not 100% correct, better say ice grain growth

l. 441: Mention this is the methods already

Fig. 9a) Think about using a perceptually uniform colour sheme than jet/rainbow, see e.g. Crameri et al. (2020) for details

l. 465 annealed samples…ice core data suggest that abnormal….

l. 466: "An annealing" or "annealing experiments of ice samples"

l. 467: Better "strengthening" than "enhancement"

Fig. 10: I like the idea of a schematic drawing to communicate the main conclusion of the large amount of data. However, I suggest to emphasize the difference between both options (e.g. by writing on the arrows). The example of minimized normal grain growth looks exactly the same as the

starting material, som slight modifications would be good to avoid the interpretation that nothing at all is happening.

Conclusions: Use full sentences, the numbering is not necessary. Mention where Priestley ice is from

l. 476: define "sampling", does it include drilling, core handling, first processing etc?

l. 477: clarify bubbles at ice grain boundaries (all, most, few)

l. 479: grain boundary. Abnormal grain growth is an additional process

l. 480: avoid generalizations: grain size change in our (natural) samples/ in the depth range investigated

l. 481: Number density of bubbles

l. 486: …constrast can drive abnormal grain growth…

l. 487: inhibits grain boundary migration of ice grains / in the ice matrix

l. 490: CPO usually gets stronger with depth, is this relevant for your comparison?

Literature

Crameri, F., Shephard, G.E. & Heron, P.J. The misuse of colour in science communication. *Nat Commun* **11**, 5444 (2020). https://doi.org/10.1038/s41467-020-19160-7

Stoll N, Eichler J, Hörhold M, Shigeyama W and Weikusat I (2021a) A Review of the Microstructural Location of Impurities in Polar Ice and Their Impacts on Deformation. Front. Earth Sci. 8:615613. doi: 10.3389/feart.2020.615613

Stoll, N., Eichler, J., Hörhold, M., Erhardt, T., Jensen, C., and Weikusat, I.: Microstructure, micro-inclusions, and mineralogy along the EGRIP ice core – Part 1: Localisation of inclusions and deformation patterns, The Cryosphere, 15, 5717–5737, https://doi.org/10.5194/tc-15-5717-2021, 2021b.

Stoll, N., Hörhold, M., Erhardt, T., Eichler, J., Jensen, C., and Weikusat, I.: Microstructure, micro-inclusions, and mineralogy along the EGRIP (East Greenland Ice Core Project) ice core – Part 2: Implications for palaeo-mineralogy, The Cryosphere, 16, 667–688, https://doi.org/10.5194/tc-16-667-2022, 2022.

Bohleber, P., Stoll, N., Rittner, M., Roman, M., Weikusat, I., & Barbante, C. (2023). Geochemical characterization of insoluble particle clusters in ice cores using two-dimensional impurity imaging. *Geochemistry, Geophysics, Geosystems*, 24, e2022GC010595. https://doi.org/10.1029/2022GC010595

Stoll, N., Westhoff, J., Bohleber, P., Svensson, A., Dahl-Jensen, D., Barbante, C., and Weikusat, I.: Chemical and visual characterisation of EGRIP glacial ice and cloudy bands within, The Cryosphere Discuss. [preprint], https://doi.org/10.5194/tc-2022-250, in review, 2023.

---

## Referee Comment (RC2)

**Review of Sheng Fan et al.'s - The Cryosphere preprint:- tc_2022-228**

**Grain growth of natural and synthetic ice at 0ºC**

**General comments on the manuscript**

This preprint by Fan et al. suggests grain growth in natural ice is much slower than synthetic ice and is based on two sets of experiments. The authors successfully obtain high-resolution EBSD data that provide insights into microstructural and CPO changes. The paper is generally well written, but suffers from some inconsistencies and repetitions and some minor rearrangement, shortening and streamlining of the text is required. There are far too many one sentence paragraphs or subsections in this manuscript that do not let the reader to obtain a good sense of flow. If substantial modifications are undertaken then it would be suitable for The Cryosphere.

**Specific comments for the author's consideration**

- This papers introduction (lines 33-44) contains general statements on the mechanical behaviour of ice which are not relevant to their annealing story. I would suggest it would be better to highlight why there is a need to better understand grain growth in ice and formulate the aims of the paper more clearly.
- In the Abstract and also in introduction it would be good to point out that these are both short- and long-term annealing experiments. In the text it would also be an idea to say how these compare with the time frame summarized in the plot (Fig. 13) in Wilson et al (2014).
- Annealing times are used inconsistently throughout the manuscript. Days are used in Fig. 2; Hours in Figs. 4 & 5; Seconds in Fig. 7; Hours in Figs 8, 9. On some of the plots it may be worth having two timescale bars included. Hours used on lines 404, and months on line 474.
- The results section is very disjointed and in need of some consolidation. E.g. Why not combine sections 3.2.1 and 3.2.2; 3.3.1 and 3.3.3 and the subheading "3.3.2 Bubbles" could be better identified.
- There is no proper introduction to the Discussion' section (4). It would be good if there is a summary that sets the scene for the following subsections. In fact much of section 4.1.1 could be removed as many of the equations have little bearing on the discussion in section 4.1.2.
- The section on evaluating the bubbles in the Priestley ice (4.2) needs to be considerably shortened and unnecessary referencing needs to be scaled back.
- The section 4.3 could also be significantly shortened and the discussion and referencing of previous work (e.g. lines 430-435) is out of place and could be deleted.
- Nowhere in this manuscript has there been a discussion of strain energy and its role as a contributor to grain boundary migration.
- Overall, this paper presents some high-quality experimental data. However, conclusions are too focused on the Priestley glacier and do not suggest the significance of this work and how it compares with Azuma's research and how it could be applied to other ice sheets or glaciers.

- **Editorial comments keyed to line numbers**

55- It would be good to see some specific references to papers that demonstration how dust and dissolved salts effect the ice microstructure.

105- I have difficulty relating text statements referring to the ice flow plane with Fig. 1a? Why isn't the flow plane clearly identified in the figure? In the natural ice was there any suggestion of a grain shape alignment parallel to the flow plane?

Table 1. - I feel the caption could be expanded to better identify what S_M_A P_B etc sand for and shorten the text between lines 155-158.

162 -  The sentence "For a few….. (Table 1" could be removed as this is already said on line 145 at bottom of Table.

185 & 195 -  Is it necessary to have (SE) in the text and caption. It would be better to write it out in full.

213 – Why write out CPO in full again when it was used on lines 44, 46, 60 etc.

221 – Why repeat the spelling out of SPO when this was undertaken on line 219? Again it is repeated on line 279, 289 (in caption) and elsewhere in the paper, e.g. line 360.

225 – 235 – These two sections should be removed from here as they repeat material that should be in the methods section. This will then require renumbering and possibly renaming the following sections.

243 – delete 'experiment'.

278 – SPO problem.

279 – Remove unnecessary parenthesis.

306 – Why is it really necessary to have subsections 4.1 and 4.1.1 ? you need to reorganize subheadings.

310-331 – Except for the first paragraph, the rest of this section does not contribute anything to this highly descriptive paper. The parameters $n, k$ and $t_0$ are not really defined. There are equations, which are really not used and their relevance is unclear. Why not explain these parameters in the figure caption to Fig. 7?

318 – change 'and others' to et al.

339-340 – Why not combine these two paragraphs and shorten them in the light of writing a new introduction to the "Discussion" section.

Fig. 7b – The significance of this figure and the rate constant, $k$, needs to be better clarified in a new introduction to the "Discussion". Or this figure could be removed.

364 – References not in reference list. Are they necessary?

365-375 Much of this discussion and references are irrelevant to the origin of bubbles, and could be deleted.

380-395 and to 425 – This section can be considerably condensed.

420 – The sentence "Further studies…natural ice" could be deleted and especially the word "elle" is not required.

485-490          – This conclusion has not been supported by any data in this manuscript.

**References**

Azuma et al., 2012, Impeding effects of airbubbles on normal grain growth, JSG 42, 184-193.

Wilson, C.J.L., Peternell, M., Piazolo, S., and Luzin, V., 2014. Microstructure and fabric development in ice: lessons learned from *in situ* experiments and implications for understanding rock evolution. *Journal of Structural Geology* **61**, 50-77. http://dx.doi.org/10.1016/j.jsg.2013.05.006.

---

## Author Comment (AC1)

<h1 style="text-align: center;">Response to Reviewer 1, Dr Nicolas Stoll</h1>

We thank reviewer 1, Dr Nicolas Stoll for his thoughtful and helpful reviews of our paper. The comments have helped us to clarify many important points. This document outlines point-by-point responses to the reviewer. The reviewer's comments are in blue type; our responses are in black type; *extracts from the revised manuscript are in italics.*

**R.1.1** Fan et al. provide an interesting and well-structured manuscript regarding the differences in grain growth of natural and synthetic ice. Despite being investigated for several decades, grain growth of ice crystals remains a topic with many open questions. This is mainly due to the challenges in observing grain growth in situ, i.e. in glaciers or ice sheets, and in setting-up laboratory experiments with boundary conditions similar to natural conditions (time, strain). The presented manuscript compares synthetic ice samples with natural samples from Antarctica and discusses the differences in the microstructure observed during the experimental time frame.
The manuscript is well-written, provides good figures, and the experiments have been carried out thoroughly. The topic is of interest for the community and the data gathering and analysis is well done as far as I can evaluate it. However, the motivation, and especially the decision to conduct experiments at 0°C, should be explained in more detail. I am not an expert in annealing studies and 0°C are usually avoided at all costs in microstructural ice core research. With some work on the issues mentioned below, I am sure that the manuscript can be published after minor revisions in The Cryosphere.

We thank the reviewer for his affirmation of our work. We agree with the reviewer that the motivation to conduct experiments at 0°C should be explained in more detail. We will add additional statements in the Introduction section to clarify this paper's motivation.

**General comments**

**R.1.2** I agree that crystal grain growth in ice is far from understood and that temperature certainly plays a role. However, the motivation of the manuscript remains a little unclear to me. You present examples (l.72-78), but these are merely very specific cases. Mechanical drilling is preferred over hot-water drilling for the majority of ice cores, especially for deep ice cores, and subglacial water inflow into a borehole is scarce and does not impact the already drilled ice, thus a maximum of a few meters. The vast majority of natural ice samples are well preserved for microstructure analysis or are discarded right away. Hence, I would encourage you to elaborate a bit more on the perks of the new insights regarding grain growth at 0°C (e.g. maybe that hot-water drilling could be used more frequently)?

We thank the reviewer for providing suggestions on clarifying the motivations. In the modified manuscript, we will add additional statements to clarify our motivations, which include (1) assess the grain growth mechanisms in natural ice at 0°C, (2) assess the impact of hot-water drilling on the modification of ice microstructure, and (3) use ice as an analogue to understand the annealing of minerals at their melting temperatures.

**R.1.3** The language is overall fine, but more (small) mistakes occur towards the end of the manuscript. I try to mention them below, but another spelling check would be beneficial.

We thank the reviewer for pointing out our writing mistakes. We will correct these mistakes accordingly (detailed are provided in the replies to the specific comments).

**R.1.4** I would be interested in a real photograph of the annealing experiment. Fig. 2 gives a good overview, but a photograph in the Appendix would be useful.

We will provide photographs of the annealing experiments in the supplementary materials.

**Specific comments**

**R1.5** l. 22: natural ice is characterized

Corrected.

**R1.6** l. 23: I think you mean "grain boundaries are pinned by bubbles"

Apologize for our vague writing. We have corrected it.

**R1.7** l. 23: l. 26: in the grain interior

Corrected.

**R1.8** l. 27: It is not totally clear what you mean by one stage, maybe add half a sentence to describe it.

We agree with the reviewer that the description of "one stage" is unclear. We have removed this sentence in the modified manuscript.

**R1.9** l. 30: the wording is unprecise, "ice grain growth" or "growing of the ice crystals" might be better.

We agree with the reviewer. We have modified the sentence to *We suggest the bubble-pinning, which inhibits grain growth,…*

**R1.10** l. 31: I would like to read one last sentence in the abstract briefly describing what your results mean.

We have added a new statement to describe the meaning of our findings to natural ice:

*The observed abnormal grain growth with a weakening of CPO intensity in our annealed samples, in good agreement with the weak CPO in the shallow part of the Priestley ice core, suggests that abnormal grain growth could be common in the shallow part of the Priestly glacier.*

**R1.11** l. 44: delete "statistics"

Corrected.

**R1.12** l. 56: add "comparably impurity-rich"; ice is still a very pure material.

Corrected.

**R1.13** l. 57: Weikusat et al., 2017b deals only very briefly with impurities and does not investigate them; for more recent work refer to e.g. Bohleber et al., 2023, Stoll et al., 2022, 2023.

We thank the reviewer for pointing out that some more recent references should be added. We have added Stoll et al. (2021) as a new reference.

**R1.14** l . 62: For recent microstructural results from a deep ice core maybe add Stoll et al., 2021b.

We have added Stoll et al. (2021).

**R1.15** l. 64: The sentence is certainly true, but it is important to add that several processes are not understood yet. Especially the relationship between grain growth, fabric/deformation, and impurities/bubbles remains unclear. Zener pinning has, to my knowledge, only been observed very rarely in ice and the presented evidence raises some doubts. You could cite Stoll et al., 2021a and references within for a recent review and overview of the topic without going into to details here (since it is not the scope of this manuscript).

We thank the reviewer for his suggestion. However, we respectfully argue that this paragraph has already clearly stated the complication of factors that affect ice grain growth (as agreed by the reviewer) and we also suggest sufficient citations have been provided.

**R1.16** l. 67: maybe replace original with host.

We respectfully suggest keeping the description of "original grains" as it is commonly used for describing nucleation processes in rock and ice deformation literature (e.g., Fan et al., 2021; Halfpenny et al., 2006).

**R1.17** l. 74: The entire ice core or "just" the outer rim? As mentioned above, hot-water ice cores are not primarily used for microstructural analysis as far as I know

Previous studies have used entire ice cores acquired from hot water drilling for microstructural studies (e.g., Jackson, 1999; Jackson & Kamb, 1997). We agree with the reviewer that our previous statement is not robust enough. Therefore, we have modified the statement to *For example, during hot-water drilling, the circulation of pressurized hot water, which is used to produce and maintain the opening of boreholes, can lead to an increase in the ice-core temperature*

**R1.18** 76 ff: Backflow of water is luckily not usual, but occurs in special occasions, such as the EPICA drilling, and is tried to be avoided. Furthermore, there are many problems if such backflow and refreezing occurs, the impacted ice will certainly not be used for "normal" microstructural analysis. Mentioning this rare occasion as a major objective is thus not very strong. Similarly, natural samples are usually well preserved or discarded before analysis. Rephrase it to "can occur" to avoid drawing a misleading image here and rethink the justification of your objectives, I am sure there are better fitting options to address.

We apologise for our misleading statements. Therefore, we have modified the statement to *Moreover, the back-flowing of subglacial water into the borehole, which might happen during the drilling of deep ice cores*

**R1.19** l. 83: Merge both sentences: ..kinetics to better understand grain growth in natural ice".

We have merged the sentences as suggested.

**R1.20** l. 92: space between of25.4

Corrected.

**R1.21** l. 96: replace eliminated with discard/vanish/exude or something similar

Corrected.

**R1.22** l. 98: ice slabs are used

Corrected.

**R1.23** l. 99: thickness of ice slabs

Corrected.

**R1.24** l. 100: is there a more specific name than Antarctic ice core no 30.?

The core number used in this paper remains consistent with the original report (Thomas et al., 2021) and we currently don't have a specific name for it.

**R1.25** l. 100: is there a more specific name than Antarctic ice core no 30.?

The core number used in this paper remains consistent with the original report (Thomas et al., 2021) and we currently don't have a specific name for it.

**R1.26** l. 104: Fig. number missing

We apologize for this mistake. The figure number has been corrected.

**R1.27** Fig. 1: I think it would be helpful to clearly define your natural ice samples once and then just refer to it, thus either natural or Priestley ice (I would suggest the latter).

We have corrected the labelling to Priestley ice.

**R1.28** l. 110: Cold laboratory

The name of our lab is the "Ice Lab". Therefore, we would like to keep the expression of "Ice Lab".

**R1.29** l. 115: the bins

We only have one chilly bin. Therefore, we would like to keep the expression of "the chilly bin".

**R1.30** l. 118: hamper/impact

Thank you for the suggestion. We have modified the statement to … *as the contamination of silicon oil on the sample surface will hamper the collection of electron backscatter diffraction (EBSD) data.*

**R1.31** l. 124: There might be more precise words than recharging and maintained. Once can be deleted.

Thank you for the suggestion. We have modified the statement to *During experiments, we stabilised the temperature of silicon oil at ~0ºC by refilling the water-ice bath every seven days.*

**R1.32** l. 142: Maybe add dewar flask/bottle for readers not acquainted with laboratory terms.

Thank you for the suggestion. We have modified the statement to … *a liquid nitrogen dewar tank.*

**R1.33** Table 1: Clarify the parameter for grain size, is it diameter? I am confused by sample 12 and 13, the ice grain numbers don´t add up as they do in the other samples. If they are not available it´s fine, but in sample 15 N/A does not have an impact on the total number. Please clarify. You usually refer to a "thin slice", here to a "thin section" – stick with one term or do they differ?

Thank you for pointing out these mistakes. We have specified that the grain size is the measurement of the area-equivalent diameter in the text. We have also added this information in Table 1 in the modified manuscript. We apologize for the wrongly recorded number of grains for samples 12_P_A, 12_P_B, 13_P_A, and 13_P_B. This mistake has been corrected in the modified manuscript. We have also modified the expression of "thin section" to "thin slice".

**R1.34** Fig. 2b): The small plot is a bit confusing. Maybe you can think of a clearer way to present it, if this information is crucial.

We have modified Fig. 2b so that the small plot is clearly labelled as *a zoomed-in view of a sealed aluminium vessel*.

**R1.35** Fig. 2b): The small plot is a bit confusing. Maybe you can think of a clearer way to present it, if this information is crucial.

We have modified Fig. 2b so that the small plot is clearly labelled as *a zoomed-in view of a sealed aluminium vessel*.

**R1.36** l. 155: space at 5mm

Corrected.

**R1.37** l. 169: Describe briefly how you prepared the slices, did you microtome the surface?

We have added a new statement to explain the sample-polish procedure: *We acquired a polished sample surface by hand lapping on grit papers with grit sizes of 80, 240, 600, 1200 and 2400 at -40°C.*

**R1.38** Fig. 3: Put the scale in panel a and e.

Corrected.

**R1.39** l. 213: CPO already introduced, the same applies for SPO later in the text.

We have corrected this mistake for the whole manuscript.

**R1.40** l. 217: Briefly introduce the M-index and add "the".

We have added "the" to the front of the "M-index". The M-index is a standard method for quantifying CPO strength. We suggest that our original text has already provided a brief and clear explanation of the M-index.

**R1.41** l. 219: …bubbles, as done for calculating grain sizes.

Corrected.

**R1.42** l. 222: delete "will"

Corrected.

**R1.43** l. 226-229: Is this needed? You explain the different aspects in more detail, I think you don´t need this introduction.

We agree with the reviewer. We have removed this paragraph.

**R1.44** l. 231-235: I think it´s important to have a temperature record to evaluate the experiments, but a  designated subsection might not be needed. Maybe you find a way to implement it in the other sections; if not leave it like it is.

Detailed temperature records have been provided in Figs. 2(c) and 2(d).

**R1.45** Section 3.2: medium is a relative term (as long as you don´t define it), I am not sure if the grain size has to be in the title

We agree with the reviewer. We have corrected this mistake.

**R1.46** l. 249: delete to. I think the use of "c-axis CPOs" is redundant, CPOs usually refer to the c-axis if not stated otherwise. You don´t show a-axes data here, so rephrase to e.g. "…and c-axis patterns are characterized by …

We have corrected this mistake.

**R1.47** Fig. 4c): The x-axis could be designed more accessible with e.g. 10^x

Corrected.

**R1.48** l. 261: small ice grains, otherwise you need something to compare it with

We have modified this sentence to *The starting material of Priestley ice is characterised by small ice grains with less irregular grain boundaries interlocking with large ice grains with more irregular grain boundaries*

**R1.49** l. 268: I suggest to clearly state the crystal orientations instead of the colours, which is confusing to readers not used to CPOs and the colour legend

This sentence describes Fig. 5(a). Therefore, it is necessary to describe what we see from the figure (Fig. 5(a)). The CPO has been described in Sect. 3.3.3.

**R1.50** l. 269: 12-18 times larger than that of/ 12-18 times the value of the median ice grain size

Corrected.

**R1.51** 276: Delete the part about abnormal bubbles if you don´t discuss it later

Corrected.

**R1.52** l. 277: space and(d)

Corrected.

**R1.53** l. 278: SPO introduced already

Corrected.

**R1.54** l. 279: the last part belongs into the discussion section

We have removed this part.

**R1.55** l. 281: increases with time

Corrected.

**R1.56** Fig.5c) The x-axis could be designed more accessible with e.g. 10^x

Corrected.

**R1.57** Is there any meaning to the black-white bars below c and e? The caption could be shortened by referring to Fig. 4.

The black-white bar separates the analyses of ice and bubbles. In the modified manuscript, we have changed the black-white bar to a solid thick line to avoid confusion. Priestly ice contains bubbles, which are illustrated as black blobs. These features, which do not exist in synthetic ice, need to be mentioned for Priestley ice. Therefore, we would suggest keeping the caption (which is not long) for clarification.

**R1.58** l. 294: CPO patterns are generally…

Corrected.

**R1.59** Fig. 6 caption: delete point, only pole figure. Thomas et al. (2021), similar with Bons and others and Azuma and others -> consistency in referencing

Corrected.

**R1.60** Fig. 7 Annealing time in s is tricky, try to stick to time formats already used (hours preferably). The same goes for grain size, so far you used microns.

We use seconds (s) for time and millimetres (mm) for grain size so that the unit for parameters (e.g., $k$) remain consistent with previous studies (e.g., Azuma et al., 2012).

**R1.61** l. 344: Fig. 7 does not compare, it displays the data which is a comparison. Rephrase.

We use modified the sentence to *Figure 7(b) displays a comparison between the modelling result and measurements…*

**R1.62** l. 347-355: Challenging to read as you refer to sections not read yet. Maybe rephrase or switch order.

We assume the reviewer refers to the sentence *We suggest that grain-growth inhibition in natural ice is controlled by both reduction of grain boundary mobility and driving force (as discussed in Sect. 4.2) and processes that are different from normal grain growth (as discussed in Sect. 4.3).*

We would like to politely suggest that this sentence is only a transitional sentence that links to the following discussions in sections 4.2 and 4.3. To reduce the confusion, we have modified the sentence to *We suggest that grain-growth inhibition in natural ice is controlled by both reduction of grain boundary mobility and driving force (as discussed in the following Sect. 4.2) and processes that are different from normal grain growth (as discussed in the following Sect. 4.3).*

**R1.63** l. 367: by the microstructure

Corrected.

**R1.64** l. 372: We compare the following microstructural differences…:
Delete "these microstructural differences include"

Corrected.

**R1.65** l. 375: Can you quantify "many more" with your data?

We have modified this sentence for clarification. *The Priestley ice is bubble rich as revealed by the SE image (Fig. 5(a)). In contrast, the synthetic ice does not contain visible bubbles (Fig. 4(a)).*

**R1.66** l. 379: Weikusat et al., 2017b deals only very briefly with impurities and does not investigate them; for more recent work refer to e.g. Bohleber et al., 2023, Stoll et al., 2022, 2023. For readability it would be good to stick with soluble or dissolved impurities. The discussion of impurities in your samples is tricky, since no data is published (yet). All natural ice contains impurities, this is thus not a necessary statement. I think you can shoren this paragraph and mention that impurtiies play a role (even though the details are unclear, see Stoll et al., 2021a), but you can´t discuss this point due to missing data (which is totally fine, investigating impurities would be another major approach). This is partially done in later sentences, could be easy to combine there

We thank the reviewer for his thoughtful comments. We have (1) added Stoll et al. (2021) as a reference, and (2) removed statements on preliminary data of impurities in Priestley ice.

**R1.67** l. 386: strong CPO pattern – which one?

We have modified the statement to *The starting material of the Priestley ice has a strong CPO whilst the synthetic has a CPO that is close to random.*

**R1.68** l. 395: To avoid confusion, clearly define that you are talking about the number to area ratio as density, and did not measure the density of the bubble.

We have modified the statement to *We separately calculated the density (i.e., number per unit area) of bubbles.*

**R1.69** l. 397: bubble density remains similar for all bubbles? Clarify that you talk about different (location) type of bubbles and maybe name them instead of using "all bubbles"

We apologize for the misleading statement. We have modified the sentence to … *the bubble density remains similar for bubbles at different locations…*

**R1.70** Fig. 8: You use bubble density in the text, but not on the respective y-axis. Explain it in the text/caption and then use it accordingly

We have corrected this mistake. Please refer to R1.68.

**R1.71** l. 405: To me, the density does not remain stable in 8d) for the first 400 hours.

We have modified this sentence to *The bubble density remains relatively stable before ~400 hours…*

**R1.72** l. 407: density of bubbles evokes material density thoughts, stick to bubble density

This statement describes the density of bubbles at different locations.

**R1.73** l. 421: give examples of previous studies or use the singular. Grain boundaries can modify.

We have modified this statement to *Azuma et al. (2012) suggest grain boundaries can…*

**R1.74** l. 422: Simply say that the applied methods cannot investigate bubble movement, that´s totally fine and does not have to be defended

We have modified this paragraph to simplify the statement.

**R1.75** l. 427: delete much and directly name the size difference

Corrected.

**R1.76** l. 430: replace production with "development" or "attribute abnormal grain growth with" -> the sentence can be shortened

Corrected.

**R1.77** l. 434: This is confusing. Grains are not really surrounded by grain boundaries, but contain planar defects/dislocations separating the material inhibiting different orientations. Give examples of these characteristics. Grain growth of the ice matrix is redundant and not 100% correct, better say ice grain growth

We have modified the statement to *...the occurrence of abnormal grain growth usually correlates with initially slow grain growth and some grain boundaries with different characteristics...*

*Grain growth of ice matrix -> ice grain growth*

**R1.78** l. 441: Mention this is the methods already

This statement was not mentioned in the Methods and we suggest it is necessary to keep it here for clarity.

**R1.79** Fig. 9a) Think about using a perceptually uniform colour sheme than jet/rainbow, see e.g. Crameri et al. (2020) for details

Corrected.

**R1.80** l. 465 annealed samples…ice core data suggest that abnormal….

Corrected.

**R1.81** l. 466: "An annealing" or "annealing experiments of ice samples"

Corrected.

**R1.82** l. 467: Better "strengthening" than "enhancement"

Corrected.

**R1.83** Fig. 10: I like the idea of a schematic drawing to communicate the main conclusion of the large amount of data. However, I suggest to emphasize the difference between both options (e.g. by writing on the arrows). The example of minimized normal grain growth

looks exactly the same as the starting material, som slight modifications would be good to avoid the interpretation that nothing at all is happening. Conclusions: Use full sentences, the numbering is not necessary. Mention where Priestley ice is from

Thank you for the suggestions. We have modified the figure accordingly.

**R1.84** l. 476: define "sampling", does it include drilling, core handling, first processing etc?

Corrected.

**R1.85** l. 477: clarify bubbles at ice grain boundaries (all, most, few)

Corrected.

**R1.86** l. 479: grain boundary. Abnormal grain growth is an additional process

Corrected.

**R1.87** l. 480: avoid generalizations: grain size change in our (natural) samples/ in the depth range investigated

Corrected.

**R1.88** l. 481: Number density of bubble

Corrected.

**R1.89** l. 486: …constrast can drive abnormal grain growth…

Corrected.

**R1.90** l. 487: inhibits grain boundary migration of ice grains / in the ice matrix

Corrected.

**R1.91** l. 490: CPO usually gets stronger with depth, is this relevant for your comparison?

We suggest the abnormally weak CPO at the shallow part of Priestly ice is tightly correlated with abnormal grain growth. Therefore, we suggest this statement is relevant to this paper.

**References**

Azuma, N., Miyakoshi, T., Yokoyama, S., & Takata, M. (2012). Impeding effect of air bubbles on normal grain growth of ice. *Journal of Structural Geology*, *42*(C), 184–193. https://doi.org/10.1016/j.jsg.2012.05.005

Fan, S., Prior, D. J., Cross, A. J., Goldsby, D. L., Hager, T. F., Negrini, M., & Qi, C. (2021). Using grain boundary irregularity to quantify dynamic recrystallization in ice. *Acta Materialia*, *209*, 116810. https://doi.org/10.1016/j.actamat.2021.116810

Halfpenny, A., Prior, D. J., & Wheeler, J. (2006). Analysis of dynamic recrystallization and nucleation in a quartzite mylonite. *Tectonophysics*, *427*(1–4), 3–14. https://doi.org/10.1016/j.tecto.2006.05.016

Jackson, M. (1999). *Dynamics of the Shear Margin of Ice Stream B, West Antarctica*. Caltech.

Jackson, Miriam, & Kamb, B. (1997). The marginal shear stress of Ice Stream B, West Antarctica. *Journal of Glaciology*, *43*(145), 415–426. https://doi.org/10.1017/S0022143000035000

Thomas, R. E., Negrini, M., Prior, D. J., Mulvaney, R., Still, H., Bowman, H., et al. (2021). Microstructure and crystallographic preferred orientations of an azimuthally oriented ice core from a lateral shear margin: Priestley Glacier, Antarctica. *Frontiers in Earth Science*, *9*(November), 1–22. https://doi.org/10.3389/feart.2021.702213

---

## Author Comment (AC2)

**Response to Reviewer 2, Prof. Christopher J. L. Wilson**

We thank reviewer 2, Prof. Christopher J. L. Wilson for his thoughtful and helpful reviews of our paper. The comments have helped us to clarify many important points. This document outlines point-by-point responses to the reviewer. The reviewer's comments are in blue; our responses are in black; *extracts from the manuscript are in italics.*

**R.2.1** This preprint by Fan et al. suggests grain growth in natural ice is much slower than synthetic ice and is based on two sets of experiments. The authors successfully obtain high-resolution EBSD data that provide insights into microstructural and CPO changes. The paper is generally well written, but suffers from some inconsistencies and repetitions and some minor rearrangement, shortening and streamlining of the text is required. There are far too many one sentence paragraphs or subsections in this manuscript that do not let the reader to obtain a good sense of flow. If substantial modifications are undertaken then it would be suitable for The Cryosphere.

We thank the reviewer for his affirmation of our work. We agree with the reviewer that the rearrangement and shortening of the text is necessary. We have modified the manuscript by following the reviewer's comments.

**Response to specific comments for the author's consideration**

**R.2.2** This papers introduction (lines 33-44) contains general statements on the mechanical behaviour of ice which are not relevant to their annealing story. I would suggest it would be better to highlight why there is a need to better understand grain growth in ice and formulate the aims of the paper more clearly.

We agree with the reviewer that lines 33-44 are less relevant to the focus of this paper. We have removed them. Moreover, we rephrased this paragraph so that one of our motivations, which is understanding grain growth is important for modelling ice-sheet dynamics, has been emphasized.

**R.2.3** In the Abstract and also in introduction it would be good to point out that these are both short- and long-term annealing experiments. In the text it would also be an idea to say how these compare with the time frame summarized in the plot (Fig. 13) in Wilson et al (2014).

We thank the reviewer for suggesting that we should (1) specify the scale of annealing time, and (2) compare it with previous studies.

We have modified a statement in the abstract *To understand better grain-growth processes and kinetics, we compared microstructural data from synthetic and natural ice samples that were annealed at ice-solidus temperature (0ºC) to successfully long durations (from a few hours to 33 days).*

We have added a new statement in the introduction *The annealing time, which expands from a few hours to 33 days, is consistent with previous studies (Wilson et al., 2014; Azuma et al., 2012).*

**R.2.4** Annealing times are used inconsistently throughout the manuscript. Days are used in Fig. 2; Hours in Figs. 4 & 5; Seconds in Fig. 7; Hours in Figs 8, 9. On some of the plots it may be worth having two timescale bars included. Hours used on lines 404, and months on line 474.

We apologize for this inconsistency. We have followed the reviewer's comments and modified the figures and text accordingly to address such inconsistency.

      1. The time unit of "month" has been replaced by "days" in the text.

      2. In Fig. 2(c), we use the unit of both hours and days in the x-axis.

      3. In Fig. 7, we use the unit of both seconds and hours in the x-axis. This is because we would like to keep the unit of k ($mm^2s^{-1}$) consistent with previous studies (e.g., Azuma et al., 2012).

**R.2.5** The results section is very disjointed and in need of some consolidation. E.g. Why not combine sections 3.2.1 and 3.2.2; 3.3.1 and 3.3.3 and the subheading "3.3.2 Bubbles" could be better identified.

Thank you for the suggestion. We have (1) merged Sects. 3.2.1 and 3.2.2, (2) merged Sects. 3.3.1 and 3.3.3 as the new Sect. 3.3.1.

**R.2.6** There is no proper introduction to the Discussion' section (4). It would be good if there is a summary that sets the scene for the following subsections. In fact much of section 4.1.1 could be removed as many of the equations have little bearing on the discussion in section 4.1.2.

We apologize for a lack of introduction at the beginning of Sect. 4. We have added the following paragraph:

*The microstructural evolution is fundamentally different between synthetic and Priestley ice during annealing (Sect. 3.2, 3.3). To explore ice-grain growth mechanisms, we start by comparing the evolution of ice-grain size between synthetic and Priestley ice (Sect. 4.1). After that, we focus on interpreting the microstructural-evolution data from Priestley ice to understand (1) the role of bubbles in the inhibition of ice-grain growth (Sect. 4.2), and (2) mechanisms that control the activation of abnormal grain growth and how does abnormal grain growth modify the grain size and CPO in natural ice (Sect. 4.3).*

We would like to politely suggest keeping section 4.1.1 and equations included in this section. Because Eqs. (1)–(5) are co-dependent, and they are essential for the audience to understand how did we calculate the value of *n* and *k,* which are the key for discussion in the Sect. 4.1.2.

**R.2.7** The section on evaluating the bubbles in the Priestley ice (4.2) needs to be considerably shortened and unnecessary referencing needs to be scaled back.

We agree with the reviewer. The following statements from the original manuscript have been removed so that this section only focuses on discussing bubbles:

*(2) Insoluble and soluble impurities. Natural ice contains insoluble impurities, such as $CaSiO_3$ and $SiO_2$, and soluble impurities, such as ions produced from dissolved salts (Baker et al., 2003; Faria et al., 2010; Gow, 1968; Gow & Williamson, 1971; Stoll et al., 2021; Svensson et al., 2005; Weikusat et al., 2017). The methods applied in this study do not enable us to locate impurities within grains and/or at grain boundaries. In contrast, the synthetic ice was produced from ultra-pure water (Sect. 2.1); therefore, the content of insoluble and soluble impurities should be negligible.*

*(3) Geometrically necessary dislocation (GND) density. The Priestley ice develops intragranular boundaries (Fig. 5(b); Sect. 3.3.1), indicating a relatively high GND density. In contrast, the synthetic ice has few intragranular boundaries (Fig. 4(b); Sect. 3.2.1), indicating a relatively low GND density.*

*Previous studies suggest grain boundary can modify the surface tension of air bubbles; consequently, bubbles can be dragged by grain boundaries and move via water-molecule transportation (Azuma et al., 2012). We did not directly observe the movement of air bubbles in this study. This is because our data are "snapshots" of ice and bubble microstructures. Understanding the kinematics of bubble movement requires additional input from in-situ annealing experiments, where bubble positions can be continuously monitored.*

We added a short statement to clarify why other possible parameters such as impurities, CPO, and strain energy are not discussed.

*In the following paragraphs, we will focus on evaluating the impact of bubbles on the inhibition of ice grain growth. Evaluating the impact of impurities, CPO, and strain energy on the grain growth of ice would require additional data input and extensive modelling that are beyond the scope of this paper.*

**R.2.8** he section 4.3 could also be significantly shortened and the discussion and referencing of previous work (e.g. lines 430-435) is out of place and could be deleted.

We agree with the reviewer. We have removed lines 430–435.

**R.2.9** Nowhere in this manuscript has there been a discussion of strain energy and its role as a contributor to grain boundary migration.

We thank the reviewer for pointing out that there is a lack of discussion of strain energy in the current manuscript. We have attempted to explore the role of dislocation density on the abnormal grain growth in Sect. 4.3 using our EBSD data. However, to further understand the role of strain energy on grain growth we would need to (1) add annealing experiments on experimentally deformed samples, and (2) add modelling that can evaluate the hypothesis derived from experiments. We are currently conducting ELLE modelling using deformed and undeformed samples as inputs. One of the motivations of this modelling project is to evaluate the role of strain energy on grain growth. Therefore, evaluating the role of strain energy on grain growth is out of the scope of this paper and it will be thoroughly discussed in our future papers.

**R.2.10** Overall, this paper presents some high-quality experimental data. However, conclusions are too focused on the Priestley glacier and do not suggest the significance of this work and how it compares with Azuma's research and how it could be applied to other ice sheets or glaciers.

We thank the reviewer for affirming our work. We agree with the reviewer that two improvements are required in the section of conclusion: (1) suggest the significance of our work to the natural ice system, (2) emphasize the comparison of our work with Azuma's work.

To address the first point, we have implemented the following changes to the statements:

*Abnormal grain growth introduces an additional grain-growth process to normal grain growth. Together, bubble pinning and abnormal grain growth govern the grain size change in natural ice samples.*

*Consequently, we speculate that grain growth in natural ice might comprise more than one stage and it should correspond to more than one set of grain-growth parameters.*

*Abnormal grain growth is observed in annealed natural ice samples.*

*Annealed natural ice samples that contain abnormally large grains exhibit a weaker CPO intensity compared with other annealed samples without abnormal grain growth.*

To address the second point, we have added new statements:

*Annealing experiments at 0ºC were conducted on synthetic, ultra-pure water ice samples, and natural, Priestley ice samples. The grain size of synthetic samples increases with annealing time, with a grain-growth exponent, n, of 2–3, consistent with Azuma et al. (2012).*

*The grain-growth rate in natural ice samples is much slower than predictions using grain-growth parameters derived from bubble-free synthetic ice (e.g., this study; Azuma et al. (2012)).*

**Response to editorial comments keyed to line numbers**

**R.2.11** 55- It would be good to see some specific references to papers that demonstration how dust and dissolved salts effect the ice microstructure.

We apologize for the lack of citation. We have added references that address the effect of dust and dissolved salts on ice microstructure, including Faria et al. (2014) and Stoll et al. (2021).

**R.2.12** 105- I have difficulty relating text statements referring to the ice flow plane with Fig. 1a? Why isn't the flow plane clearly identified in the figure? In the natural ice was there any suggestion of a grain shape alignment parallel to the flow plane?

The flow plane could not be clearly identified during the sample cutting in this study. Therefore, we did not mark the flow plane in Fig. 1(a). After the collection and processing of EBSD data, we could reveal grain shape and CPO, which can be used to identify the flow plane (Thomas et al., 2021).

**R.2.13** Table 1. - I feel the caption could be expanded to better identify what S_M_A P_B etc sand for and shorten the text between lines 155-158.

We have added a caption for Table 1 to explain the meaning of sample numbers:

² *The first number refers to the number of ice slab cut from ice cores. The "S" refers to synthetic ice; "M" refers to the medium grain size of synthetic ice; "P" refers to Priestley ice; "A" and "B" refers to thin slices subsampled from each ice slab.*

We would suggest to keep lines 155-158 so that the meaning of "ice slab" and "ice slice" can be clarified.

**R.2.14** 162 -  The sentence "For a few….. (Table 1" could be removed as this is already said on line 145 at bottom of Table..

We have removed this statement.

**R.2.15** 185 & 195 -  Is it necessary to have (SE) in the text and caption. It would be better to write it out in full.

Corrected.

**R.2.16** 213 – Why write out CPO in full again when it was used on lines 44, 46, 60 etc.

Apologize, we have corrected this mistake.

**R.2.17** 221 – Why repeat the spelling out of SPO when this was undertaken on line 219? Again it is repeated on line 279, 289 (in caption) and elsewhere in the paper, e.g. line 360.

Apologize, we have corrected this mistake.

**R.2.18** 225 – 235 – These two sections should be removed from here as they repeat material that should be in the methods section. This will then require renumbering and possibly renaming the following sections.

We have removed the introduction for the Sect. Result. We merged the section of temperature fluctuation to the section method.

**R.2.19** 243 – delete 'experiment'.

Corrected.

**R.2.20** 278 – SPO problem.

Corrected.

**R.2.21** 279 – Remove unnecessary parenthesis.

Corrected.

**R.2.22** 306 – Why is it really necessary to have subsections 4.1 and 4.1.1 ? you need to reorganize subheadings.

We would like to politely suggest keeping subsections 4.1.1 and 4.1.2. This is because section 4.1 discusses the grain size evolution in synthetic and natural ice. For the clarity, it would be

great to separately discuss synthetic ice and natural ice. We understand that the subheadings are unnecessarily long and we have shortened them to "*4.1.1 Synthetic ice*" and "*4.1.2 Priestley ice*".

**R.2.23** 310-331 – Except for the first paragraph, the rest of this section does not contribute anything to this highly descriptive paper. The parameters n, k and t0 are not really defined. There are equations, which are really not used and their relevance is unclear. Why not explain these parameters in the figure caption to Fig. 7?

Please refer to R.2.6 for the response.

**R.2.24** 318 – change 'and others' to et al.

Corrected.

**R.2.25** 339-340 – Why not combine these two paragraphs and shorten them in the light of writing a new introduction to the "Discussion" section.

We have added an introduction to the Sect. Discussion. For details, please refer to R.2.6.

**R.2.26** Fig. 7b – The significance of this figure and the rate constant, k, needs to be better clarified in a new introduction to the "Discussion". Or this figure could be removed.

The significance of Fig. 7(b) has been discussed in Sect. 4.1.2 as it is a key part for us to explore the $k$ value at a fixed $n$ value for Priestley ice. We would like to politely suggest keeping this figure as it is a key part of our discussion.

**R.2.27** 364 – References not in reference list. Are they necessary?

We apologize for the missing references. We have corrected this mistake.

**R.2.28** 365-375 Much of this discussion and references are irrelevant to the origin of bubbles, and could be deleted.

We agree with the reviewer that these discussions are not relevant to bubbles. However, we should suggest that it is also important to list other possibilities that can contribute to the reduction of grain-growth rate, such as second-phase pinning, CPO, and dislocation densities. We apologize that our idea was not introduced properly and thus caused confusion. We have thus added a statement to clarify this idea: *In the following paragraphs, we will focus on evaluating the impact of bubbles on the inhibition of ice grain growth. Evaluating the impact of impurities, CPO, and strain energy on ice-grain growth would require additional data input and extensive modelling that are beyond the scope of this paper.*

**R.2.29** 380-395 and to 425 – This section can be considerably condensed.

We agree with the reviewer, and we have corrected this mistake. Please refer to R.2.7 for details.

**R.2.30** 420 – The sentence "Further studies…natural ice" could be deleted and especially the word "elle" is not required.

We have removed this statement.

**R.2.31** 485-490 – This conclusion has not been supported by any data in this manuscript.

This point of conclusion is correlated with our discussion in Sect. 4.3 about dislocation densities (l 437-449 in the original manuscript). We would like to politely suggest keeping this point as it is a key part of the conclusions.

**References**

Azuma, N., Miyakoshi, T., Yokoyama, S., & Takata, M. (2012). Impeding effect of air bubbles on normal grain growth of ice. *Journal of Structural Geology*, *42*(C), 184–193. https://doi.org/10.1016/j.jsg.2012.05.005

Baker, I., Cullen, D., & Iliescu, D. (2003, January). The microstructural location of impurities in ice. *Canadian Journal of Physics*. https://doi.org/10.1139/p03-030

Faria, S. H., Freitag, J., & Kipfstuhl, S. (2010). Polar ice structure and the integrity of ice-core paleoclimate records. *Quaternary Science Reviews*, *29*(1–2), 338–351. https://doi.org/10.1016/j.quascirev.2009.10.016

Faria, S. H., Weikusat, I., & Azuma, N. (2014). The microstructure of polar ice. Part I: Highlights from ice core research. *Journal of Structural Geology*, *61*, 2–20. https://doi.org/10.1016/j.jsg.2013.09.010

Gow, A. J. (1968). Bubbles and Bubble Pressures in Antarctic Glacier Ice. *Journal of Glaciology*, *7*(50), 167–182. https://doi.org/10.3189/s0022143000030975

Gow, A. J., & Williamson, T. (1971). Volcanic ash in the Antarctic ice sheet and its possible climatic implications. *Earth and Planetary Science Letters*, *13*(1), 210–218. https://doi.org/10.1016/0012-821X(71)90126-9

Stoll, N., Eichler, J., Hörhold, M., Erhardt, T., Jensen, C., & Weikusat, I. (2021). Microstructure, micro-inclusions, and mineralogy along the EGRIP ice core - Part 1: Localisation of inclusions and deformation patterns. *Cryosphere*, *15*(12), 5717–5737. https://doi.org/10.5194/tc-15-5717-2021

Svensson, A., Nielsen, S. W., Kipfstuhl, S., Johnsen, S. J., Steffensen, J. P., Bigler, M., et al. (2005). Visual stratigraphy of the North Greenland Ice Core Project (NorthGRIP) ice core during the last glacial period. *Journal of Geophysical Research D: Atmospheres*, *110*(2), 1–11. https://doi.org/10.1029/2004JD005134

Thomas, R. E., Negrini, M., Prior, D. J., Mulvaney, R., Still, H., Bowman, H., et al. (2021). Microstructure and crystallographic preferred orientations of an azimuthally oriented ice core from a lateral shear margin: Priestley Glacier, Antarctica. *Frontiers in Earth Science*, *9*(November), 1–22. https://doi.org/10.3389/feart.2021.702213

Weikusat, I., Jansen, D., Binder, T., Eichler, J., Faria, S. H., Wilhelms, F., et al. (2017). Physical analysis of an Antarctic ice core-towards an integration of micro-and macrodynamics of polar ice. *Philosophical Transactions of the Royal Society A: Mathematical, Physical and Engineering Sciences*, *375*(2086). https://doi.org/10.1098/rsta.2015.0347

---

## Author Response (AR2)

**Response to the editor, Prof. Kaitlin Keegan**

We would like to thank Prof. Kaitlin Keegan for her thoughtful and helpful reviews of our paper. The comments have helped us to clarify many important points. Editor's comments are in blue; our responses are in black. *Extracts from the revised manuscript are in italics.*

**R.1.1** Response R 1.2: Description of '(3) use ice as an analogue to understand the annealing of minerals at their melting temperatures.', as mentioned in the response to referees, is missing in the manuscript.

We apologise for the mistake. We have made five rounds of revisions before submitting our manuscript for a second review. Describing the use of ice as an analogue for other minerals has been included in the first round of revision. However, this specific part was removed before the final revision, but we forgot to update the document of 'response to reviewer'. The reasons for us to remove the description of using ice as an analogue for other minerals are:

1. The original motivation of this study is to assess the growth rate of natural ice at extreme conditions. The data that come out from this study is expected to be used for assessing the dynamics of ice flows at where melting is common.

2. Using the grain growth of data of ice to understand the grain growth of rock-forming minerals is a good idea. However, thoroughly discussing this topic might require another paper, which should rigorously compare the microstructural data between ice and other minerals, and it is beyond this paper's scope.

**R.1.2** Line 71: remove one of the 'and's, for example: '…at the base of many glaciers and ice sheets (Schmidt et al., 2023; Davis et al., 2023) as well as ice shelves (Schodlok et al., 2016, Pritchard et al., 2012), and is predicted…'

We apologise for the mistake. We have corrected the writing.

**R.1.3** Line 141: should be '… 'A' and 'B' refer to…'

We apologise for the mistake. We have corrected the writing.

**R.1.4** Line 420: should be 'field campaigns' instead of 'field strips'

We apologise for the mistake. We have modified the sentence to:

*Temperature records that span from January 2020 to November 2022 show…*

**R.1.5** Table 1: The last row on Page 6 is missing the Sample Type (sample 12_P_A); the first row on Page 7 is missing the Annealing Time, Initial Median Grain Size, Measuring Grain Size – Combined Sections, Number of Ice Grains – Combined Sections (sample 12_P_B). If that's intentional, place a '-' or 'N/A' to indicate that these blank boxes are intentional.

We suggest this should be a display error while the Word software tries to display vertically merged cells. We have applied modification so that the information of 12_P_A and 12_P_B can be correctly displayed.

**R.1.6** Section 2.4.3: I respectfully disagree that the broad readership of The Cryosphere will understand what the M-index is from what is included in this Methods section. As referee Stoll points out, it would be useful to include how the M-index calculates CPO intensity for the non-expert reader.

We apologise that our previous reply was not considering enough. We have added the following statement to describe the calculation of M-index:

*The calculation of M-index is based on the distribution of misorientation angles calculated from random pairs of pixels indexed as ice from a given EBSD map (Skemer et al., 2005).*

**R.1.7** Section 3.3: You refer to a Section 3.3 in your response to referee Stoll (R 1.49), referee Wilson (R 2.5), and on lines 250, 285, 356, 358, 390, 395, 396, and 418, but it does not exist. Please check your Section numbering references throughout the text.

We apologise for this mistake. We merged sections based on comments from Prof. Chris Wilson. In detail, we have (1) removed Sect. 3.2.1, (2) merged Sects. 3.2.1 and 3.2.2 as the new Sect. 3.1, and (3) merged Sects. 3.3.1 and 3.3.3 as the new Sect. 3.2.1, and (3) changed Sect. 3.3.2 as the new Sect. 3.2.2.

However, we forgot to update the referenced section number in the second submission. The previous section 3.3 should be the current section of 3.2. We have corrected this mistake throughout the whole manuscript.

**R.1.8** R 1.67 – this modified statement is not present in the present version of the manuscript.

We apologise for this mistake. These statements were firstly added based on Dr Stoll's comments. However, we removed statements, including the statement on the impact of CPO on grain growth, after accepting Prof. Wilson's comments. This is because our data is insufficient to evaluate the impact of CPO on grain growth; discussing the impact of CPO can be misleading. We have honestly addressed such shortage in the modified manuscript:

*Evaluating the impact of impurities, CPO, and strain energy on grain growth would require additional data input and extensive modelling that are beyond the scope of this paper. In the following paragraphs, we will focus on evaluating the impact of bubbles on the inhibition of grain growth.*

Unfortunately, we forgot to update the change, i.e., remove the statement about the comparison of CPO patterns between synthetic and Priestly ice, in the reply to the first reviewer, Dr Stoll.

**R.1.9** R 1.71 – I agree that the bubble density does not appear to be 'relatively stable' in the data presented in Figure 8d. Explaining the jump from time 0 to 100 hours for each bubble size, and the difference in trend between the largest bubbles (orange squares) and the smaller bubbles would be helpful.

We apologize that our previous interpretation of the evolution of bubble statistics is not robust enough. We have thus separately described and interpreted the evolution of statistics for bubbles on grain boundaries and bubbles within grains:

*For bubbles within ice grains, the density of relatively bigger bubbles (bubble size ≥300 μm) increases with time, whilst the density of relatively smaller bubbles (bubble size <300 μm) remains relatively stable during ~800 hours of annealing (square marks, Fig. 8(c)). This observation indicates the growth of some of the bubbles, probably driven by surface energy. Before ~400 hours of annealing, the density of bubbles on grain boundaries gradually increases (triangle marks, Fig. 8(d)). This observation suggests that more bubbles pin at grain boundaries probably during the migration of grain boundaries. By ~800 hours of annealing, the density of bubbles on grain boundaries has decreased (triangle marks, Fig. 8(d)). This observation suggests that some grain boundaries have swept through bubbles.*

To match the sequence of the description of figures in the text, we also switched Fig. 8(c) and 8(d), i.e., the previous Fig. 8(c) is the current Fig. 8(d); the previous Fig. 8(d) is the current Fig. 8(c).